# Lack of activity of recombinant HIF prolyl hydroxylases (PHDs) on reported non-HIF substrates

Matthew E Cockman[1†]*, Kerstin Lippl[2†], Ya-Min Tian[3†], Hamish B Pegg[1], William D Figg Jnr[2], Martine I Abboud[2], Raphael Heilig[4], Roman Fischer[4], Johanna Myllyharju[5‡], Christopher J Schofield[2‡], Peter J Ratcliffe[1,3‡]*

[1]The Francis Crick Institute, London, United Kingdom; [2]Chemistry Research Laboratory, Department of Chemistry, University of Oxford, Oxford, United Kingdom; [3]Ludwig Institute for Cancer Research, Nuffield Department of Clinical Medicine, University of Oxford, Oxford, United Kingdom; [4]Target Discovery Institute, Nuffield Department of Clinical Medicine, University of Oxford, Oxford, United Kingdom; [5]Oulu Center for Cell-Matrix Research, Biocenter Oulu and Faculty of Biochemistry and Molecular Medicine, University of Oulu, Oulu, Finland

**Abstract** Human and other animal cells deploy three closely related dioxygenases (PHD 1, 2 and 3) to signal oxygen levels by catalysing oxygen regulated prolyl hydroxylation of the transcription factor HIF. The discovery of the HIF prolyl-hydroxylase (PHD) enzymes as oxygen sensors raises a key question as to the existence and nature of non-HIF substrates, potentially transducing other biological responses to hypoxia. Over 20 such substrates are reported. We therefore sought to characterise their reactivity with recombinant PHD enzymes. Unexpectedly, we did not detect prolyl-hydroxylase activity on any reported non-HIF protein or peptide, using conditions supporting robust HIF-α hydroxylation. We cannot exclude PHD-catalysed prolyl hydroxylation occurring under conditions other than those we have examined. However, our findings using recombinant enzymes provide no support for the wide range of non-HIF PHD substrates that have been reported.
DOI: https://doi.org/10.7554/eLife.46490.001

*For correspondence:
matthew.cockman@crick.ac.uk
(MEC);
peter.ratcliffe@ndm.ox.ac.uk (PJR)

†These authors contributed equally to this work
‡These authors also contributed equally to this work

## Introduction

The HIF prolyl hydroxylases PHD1, 2 and 3 (or EGLN2, 1 and 3) are closely related isoforms of enzymes within the 2-oxoglutarate-dependent dioxygenase family that signal oxygen levels in human and animals cells. The enzymes catalyse the post-translational hydroxylation of specific prolyl residues in the transcription factor Hypoxia Inducible Factor (HIF)-α subunits. Prolyl hydroxylation promotes the association of HIF-α with the von-Hippel-Lindau ubiquitin E3 ligase (pVHL) and subsequent degradation by the ubiquitin-proteasome pathway. PHD-catalysed hydroxylation is highly sensitive to the availability of oxygen and provides an 'oxygen-sensing' mechanism that, via HIF, regulates a wide range of cellular and systemic responses to oxygen, including those (e.g. blood haematocrit) that operate precisely around the 'set-point' of physiological oxygen availability (for reviews, see *Schofield and Ratcliffe, 2004*; *Kaelin and Ratcliffe, 2008*; *Myllyharju, 2013*; *Semenza, 2012*).

The discovery of the PHDs raised the question as to whether there are other dioxygenase-substrate combinations with analogous oxygen sensing roles in human cells, and the assignment of additional (non-HIF) substrates of the PHDs has been a major focus of research in the field. More than 20 potential non-HIF substrates have been reported to date (*German et al., 2016*; *Luo et al., 2014*; *Xie et al., 2009*; *Guo et al., 2016*; *Köditz et al., 2007*; *Moser et al., 2015*; *Moser et al., 2013*;

*Moore et al., 2015*; *Heir et al., 2016*; *Segura et al., 2016*; *Zheng et al., 2014*; *Cummins et al., 2006*; *Rodriguez et al., 2016*; *Lee et al., 2015*; *Huo et al., 2012*; *Luo et al., 2011*; *Mikhaylova et al., 2008*; *Di Conza et al., 2017*; *Anderson et al., 2011*; *Xie et al., 2012*; *Xie et al., 2015*; *Ullah et al., 2017*; *Rodriguez et al., 2018*; *Takahashi et al., 2011*) (*Table 1*).

The existence of multiple non-HIF substrates has a number of important implications. If this were the case, it would be predicted: (i) that many non-HIF dependent biological systems would be regulated by oxygen, (ii) that PHD substrate competition might modulate the responses to hypoxia mediated by HIF, (iii) that inhibitors of the PHDs, including those in clinical trials (for review see *Maxwell and Eckardt, 2016*; *Gupta and Wish, 2017*), would have multiple consequences unrelated to HIF. We therefore sought to test reported non-HIF peptide and polypeptide substrates of PHDs in assays of prolyl hydroxylation.

Proposed substrates were tested for their ability to support prolyl hydroxylation, using both synthetic peptides representing the proposed hydroxylation site, and using predominantly full-length polypeptides/proteins prepared by coupled in vitro transcription-translation (IVTT). Direct evidence of prolyl hydroxylation catalysed by recombinant PHD enzymes was sought using a range of mass spectrometry (MS) methods. These assays were complemented by assays for hydroxyproline production using a radiochemical method (*Koivunen et al., 2006*; *Juva and Prockop, 1966*).

Unexpectedly, we did not detect enzyme-catalysed prolyl hydroxylation for any of the reported non-HIF substrates under conditions in which robust HIF prolyl hydroxylation was observed. We did observe a large number of peptide and polypeptide oxidations, but found no evidence that these were dependent on the catalytic activity of PHDs.

## Results

### Assays of prolyl hydroxylation using peptide substrates

Following reports of non-HIF substrates of the HIF prolyl hydroxylases (PHDs), we sought to measure the hydroxylation of synthetic peptides representing the reported sites of hydroxylation using in vitro assays of PHD activity. Assays were initially performed in response to particular published reports, and deployed a range of recombinant enzyme preparations, comprising either the full-length polypeptide or the active catalytic domain of the relevant PHD enzyme. These assays did not reveal hydroxylation at detectable levels on a range of reported non-HIF substrates.

We therefore decided to perform a systematic analysis of PHD-catalysed hydroxylation across all reported non-HIF substrates for which target residue(s) had been defined (*Table 1*). In this series of experiments, full-length recombinant proteins representing each of the PHD enzymes were used; PHD1 and PHD2 from an insect cell baculovirus expression system (*Hirsilä et al., 2005*) and PHD3 from *Escherichia coli* or insect cells. The enzymes were reacted with HIF-$\alpha$ peptides and those representing each of the reported sites of hydroxylation. Peptide products were analysed by matrix-assisted laser desorption/ionisation time-of-flight mass spectrometry (MALDI-TOF-MS) and electrospray-ionisation liquid chromatography-mass spectrometry (ESI-LC-MS). Based on structural and kinetic data for the PHD-HIF interaction (*Hirsilä et al., 2003*; *Chowdhury et al., 2009*), peptides were typically synthesised as 21–25 mers placing the target prolyl residues centrally within the sequence, except when hydroxylation of a specific isolated peptide had been reported, in which case this exact sequence was used instead, or in addition. In some cases, peptides representing different isoforms of the reported non-HIF substrates were also tested. Peptide sequences are listed in *Table 1—source data 1*.

A total of 44 non-HIF peptides representing putative sites of prolyl hydroxylation within 23 reported protein substrates were tested in this way. Reactions were conducted in batches, with each batch containing a parallel reaction with a HIF-1$\alpha$ peptide (human HIF-1$\alpha$: 556–574) that is known to be hydroxylated by all three PHD enzymes. Reaction products were analysed initially by MALDI-TOF-MS and subsequently by ESI-LC-MS. Each PHD isoform catalysed near complete hydroxylation of the positive control HIF-1$\alpha$ peptide. By contrast, no PHD isoform catalysed detectable hydroxylation of any other peptide. Similar results were obtained by MALDI-TOF-MS and by ESI-LC-MS. The signal-to-noise ratio was generally better with ESI-LC-MS; the results for these assays are exemplified in *Figure 1* and presented in full in *Figure 1—figure supplement 1*. Inspection of the MS spectra revealed apparent oxidation (i.e. a +16 Da mass shift relative to the unmodified substrate) on certain

**Table 1.** Non-HIF substrates tested in assays of PHD-catalysed hydroxylation.
Potential target proline residues in the proposed substrate (Gene ID, column 1) have been defined according to the sequence numbering of the canonical proteoform (Uniprot Accession, column 2).

| Substrate | Uniprot Acc # | Target site(s) | PHD isoform | Reference |
| --- | --- | --- | --- | --- |
| ACACB | O00763-1 | P343; P450; P2131 | PHD3 | *German et al., 2016* |
| ACTB | P60709-1 | P307; P322 | PHD3 | *Luo et al., 2014* |
| ADRB2 | P07550-1 | P382; P395 | PHD3 | *Xie et al., 2009* |
| AKT1 | P31749-1 | P125; P313; P318; P423 | PHD2 | *Guo et al., 2016* |
| ATF4 | P18848-1 | P156; P162; P164; P167; P174 | PHD3 | *Köditz et al., 2007* |
| CENPN | Q96H22-1 | P311 | PHD2 | *Moser et al., 2015* |
| CEP192 | Q8TEP8-3 | P2313 | PHD1 | *Moser et al., 2013* |
| EEF2K | O00418-1 | P98 | Not defined | *Moore et al., 2015* |
| EPOR | P19235-1 | P443; P450 | PHD3 | *Heir et al., 2016* |
| FLNA | P21333-1 | P2317; P2324 | PHD2 | *Segura et al., 2016* |
| FOXO3 | O43524-1 | P426; P437 | PHD1 | *Zheng et al., 2014* |
| IKBKB | O14920-1 | P191 | PHD1 | *Cummins et al., 2006* |
| MAPK6 | Q16659-1 | P25 | PHD3 | *Rodriguez et al., 2016* |
| NDRG3 | Q9UGV2-1 | P294 | PHD2 | *Lee et al., 2015* |
| PDE4D | Q08499-1 | P29; P382; P419 | PHD2 | *Huo et al., 2012* |
| PKM | P14618-1 | P403; P408 | PHD3 | *Luo et al., 2011* |
| POLR2A | P24928-1 | P1465 | PHD1 | *Mikhaylova et al., 2008* |
| PPP2R2A | P63151-1 | P319 | PHD2 | *Di Conza et al., 2017* |
| SPRY2 | O43597-1 | P18; P144; P160 | PHD1, 2, 3 | *Anderson et al., 2011* |
| TELO2 | Q9Y4R8-1 | P374; P419; P422 | PHD3 | *Xie et al., 2012* |
| THRA | P10827-1 | P160; P162 | PHD2, 3 | *Xie et al., 2015* |
| TP53 | P04637-1 | P142 | PHD1 | *Ullah et al., 2017* |
| TP53 | P04637-1 | P359 | PHD3 | *Rodriguez et al., 2018* |
| TRPA1 | O75762-1 | P394 | PHD2 | *Takahashi et al., 2011* |

DOI: https://doi.org/10.7554/eLife.46490.002

The following source data is available for Table 1:
Source data 1. Synthetic peptides tested in assays of PHD-catalysed hydroxylation.
Reported prolyl hydroxylation sites are indicated in red.
DOI: https://doi.org/10.7554/eLife.46490.003
Source data 2. Secondary structure comparison of HIF and non-HIF PHD substrates using crystallographic data and PSIPRED prediction software.
The secondary structures of metazoan HIF-α (upper panel) and reported non-HIF PHD substrates (human; lower panel) were predicted by PSIPRED (*Jones, 1999*) and, where possible, referenced to crystallographic data from the protein data bank (PDB). Predicted structural elements are defined as alpha-helical (red), beta-strand (blue), or coiled/no secondary structure (uncoloured). Note, PSIPRED does not define detailed secondary structures, such as bends/turns (green) and beta-bridges (start of a strand; yellow). Input sequences for PSIPRED were 30-mer in length with the target proline (bold) sited centrally. To limit duplication, for sequences containing multiple target residues in close proximity (i.e., less than five residues apart), only one sequence corresponding to the N-terminal target proline is shown. Metazoan HIF sequences which support human PHD2 catalytic activity in vitro are included (*Loenarz et al., 2011*): dr, *Danio rerio*; bf, *Branchiostoma floridae*; sp, *Strongylocentrotus purpurtas*; mm, *Mus musculus*; nv, *Nasonia vitripensis*; ta, *Trichoplax adhaerens*. Italicised PDB codes indicate substrates crystalized in complex with a PHD; '-' denotes end of resolved structure.
DOI: https://doi.org/10.7554/eLife.46490.004

peptides, for example ACTB/310–334 (*Figure 1*). However, in no case was an increase in the apparent oxidation detectable in reactions containing PHD enzymes, when compared with control reactions without enzyme. These enzyme independent oxidations were not analysed further in this series of experiments. Thus, these peptide-based assays did not provide any evidence for PHD-catalysed prolyl hydroxylation, within the limits of detection, across a wide range of reported sites in non-HIF proteins.

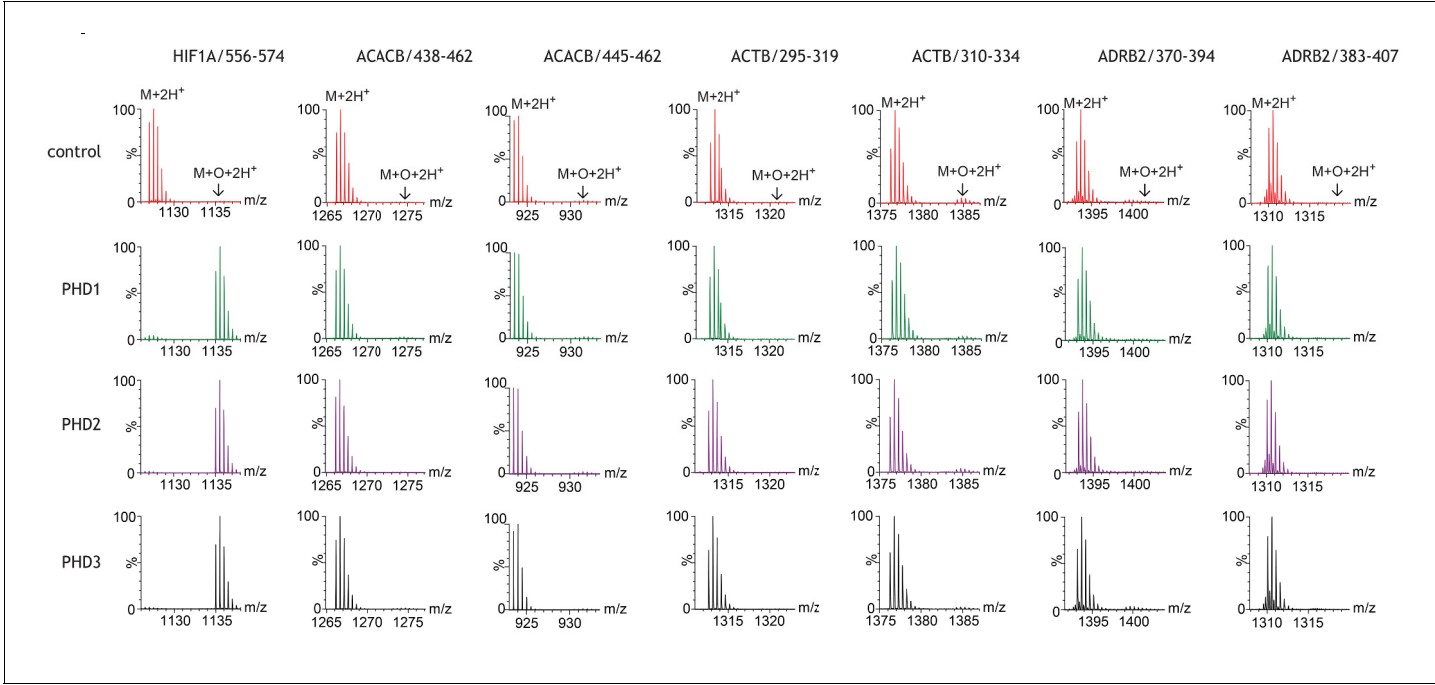

**Figure 1.** Assays of peptide hydroxylation. LC-MS spectra of peptides derived from HIF-1α (left) and selected non-HIF peptidyl substrates (see *Figure 1—figure supplement 1* for complete dataset) reacted with the indicated PHD isoform, or no PHD enzyme (control). In control reactions the doubly-charged (M+2H⁺) peptides showed the calculated mass. Following incubation with PHDs, only the doubly-charged HIF-1α peptide mass is shifted by an m/z increment of 7.997 Da (M+O+2H⁺) indicative of prolyl hydroxylation. No PHD-dependent mass shift was observed on any of the non-HIF substrates.

DOI: https://doi.org/10.7554/eLife.46490.005

The following figure supplement is available for figure 1:

**Figure supplement 1.** Assays of peptide hydroxylation.

DOI: https://doi.org/10.7554/eLife.46490.006

## Assays of prolyl hydroxylation on full-length polypeptide substrates

In many cases, the exact peptide sequence of the proposed non-HIF substrate, as opposed to the target prolyl residue in the protein, had not been reported. It therefore remained possible that more extensive sequences are required to support hydroxylation than had been presented by the peptides that we tested.

To address this, we proceeded to produce the reported PHD substrates as extended FLAG-tagged polypeptides, principally in full-length form using IVTT, and to react the IVTT products with recombinant PHD enzymes. In this series of experiments, we tested every reported enzyme-substrate couple (i.e., the proposed substrate or substrates together with the specific PHD or PHDs that had been reported to catalyse that particular hydroxylation, see *Table 2*), with the exception of RNA Polymerase II subunit RPB1, which we were unable to produce by IVTT. To ensure comparable enzyme/substrate ratios in the reactions, the soluble products of the IVTTs were quantified and normalised by FLAG-tag immunoblotting prior to reaction with the relevant PHD enzyme. The reacted IVTT products were purified by FLAG affinity chromatography, then subjected to digestion with trypsin or other proteases, as indicated in *Table 2*, and analysed by LC-MSMS. To facilitate accurate quantitation of HIF-1α hydroxylation by MS, two methionine residues were mutated to alanine within the tryptic fragment that contains the target residue Pro564 (since methionine residues are prone to non-enzymatic oxidation); these substitutions being known not to substantially alter rates of HIF-1α hydroxylation (*Tian et al., 2011*).

Reactions with non-HIF substrates were conducted in batches, in parallel with HIF-1α, to ensure activity of the PHDs under the test condition. In these assays, to increase potential hydroxylation of non-HIF substrates, the quantity of PHD enzyme was increased 2-fold or 5-fold relative to that used in the parallel HIF-1α reaction.

**Table 2.** Summary of oxidations observed on protein substrates produced by IVTT and reacted with the indicated PHD enzyme(s).

Mass spectrometry was performed on IVTT-derived substrates (column 1) reacted in the presence or absence of the indicated PHD isoform (column 2). Substrates were immunopurified by FLAG affinity and digested with the specified protease(s) (column 3) to yield peptides encompassing the putative target prolyl site, which are indicated in bold (column 4); note, during sample processing peptides containing cysteine residues were derivatized with iodoacetamide to give carbamidomethylated cysteine (+57.02). Reference to primary data for non-HIF substrates (i.e., MSMS assignment and quantitation) is provided in column five and summarised in columns 6–10. Assigned oxidations are listed in column 6; no oxidation detected (ND); oxidation detected but not localised to a specific residue (NL). Note, prolyl hydroxylation was not detected. Quantitative data for control and PHD-reacted IVTTs is given for these oxidations (columns 7–8, respectively). Low abundance peptide ions of compatible mass for oxidation, which were below the threshold for MSMS determination but present in LC-MS data (within ~5 min retention time window), were also quantified (columns 9–10). The abundance of each assigned or putative oxidation is expressed as a percentage of the non-hydroxylated peptide. Summary results for the products of control reactions that were conducted in parallel on HIF-1α to verify PHD enzyme activity are shown in columns 11–14; values were obtained by similar methods (see **Supplementary file 2** for figures depicting these primary data). Each row relates a specific 'test' reaction to its batch control HIF-1α reaction; note that high levels of activity on HIF-1α substrates were observed upon addition of exogenous PHD enzyme in all controls.

| Substrate | PHD Isoform | Protease | Quantified peptide | Figure | Oxidation | Assigned ox (%) Control | Assigned ox (%) Enzyme | Unassigned ions (%) Control | Unassigned ions (%) Enzyme | HIF-1α P402ox (%) Control | HIF-1α P402ox (%) Enzyme | HIF-1α P564ox (%) Control | HIF-1α P564ox (%) Enzyme |
|---|---|---|---|---|---|---|---|---|---|---|---|---|---|
| | | | | | | **Non-HIF substrate** | | | | **HIF-1α control** | | | |
| ACACB | PHD3 | LysC | RIPVQAVWAGWGHASENPKLPELLC(+57.02)K | 3, s1 | ND | - | - | 0.4% | 1.7% | 58% | 52% | 53% | 72% |
| | | | DVDEGLEAAERIGFPLMIK | 3, s2 | M452 | 76% | 77% | 2.3% | 0.9% | | | | |
| ACTB | PHD3 | Trypsin | DLYANTVLSGGTTMYPGIADR | 3, s3 | M305 | 71% | 48% | ND | ND | 6.3% | 9.9% | 29% | 91% |
| | | | EITALAPSTMK | 3, s4 | M325 | 54% | 30% | 0.3% | 0.1% | | | | |
| ADRB2 | PHD3 | Trypsin | LLC(+57.02)EDLPGTEDFVGHGQGTVPSDNIDSQGR | 3, s5 | D380 | 2.8% | 1.5% | ND | <0.1% | 6.3% | 9.9% | 29% | 91% |
| AKT1 | PHD2 | Trypsin | SGSPSDNSGAEEMEVSLAK | 3, s6 | M134 | 35% | 39% | ND | ND | 18% | 94% | 30% | 96% |
| | | | TFC(+57.02)GTPEYLAPEVLEDNDYGR | 3, s7 | ox: NL | 0.6% | 0.8% | ND | ND | | | | |
| | | | LSPPFKPQVTSETDTR | 3, s8 | ND | - | - | ND | ND | | | | |
| ATF4 | PHD3 | Elastase | GHLPESLTKPDQVAPFTFLQPLPSPG | 3, s9 | ND | - | - | ND | ND | 6.3% | 9.9% | 29% | 91% |
| | | | STPDHSFSLELGSEVDITEGDRKPDYT | 3, s10 | ND | - | - | ND | ND | | | | |
| CENPN | PHD2 | Trypsin | SLAPAGIADAPLSPLLTC(+57.02)IPNKR | 3, s11 | ND | - | - | ND | ND | ND | ND | 22% | 94% |
| CEP192 | PHD1 | Trypsin | WHLSSLAPPVVK | 3, s12 | ND | - | - | <0.1% | ND | 51% | 95% | 60% | 82% |
| EEF2K | PHD2 | Trypsin | HMPDPWAEFHLEDIATER | 3, s13 | M95 | 48% | 38% | ND | 0.2% | 0.5% | 88% | 36% | 93% |
| EPOR | PHD3 | LysC + GluC | YTILDPSSQLLRPWTLC(+57.02)PELPPTPPHLK | 3, s14 | ox: NL | 1.4% | 1.4% | ND | ND | 9.5% | 15% | 23% | 94% |
| | | | | | diox: W439 | 2.0% | 1.8% | | | | | | |
| FLNA | PHD2 | Trypsin | FNEEHIPDSPFVVPVASPSGDAR | 3, s15 | ND | - | - | 0.1% | 0.3% | ND | ND | 22% | 94% |
| FOXO3 | PHD1 | Trypsin | GSGLGSPTSSFNSTVFGPSSLNSLR | 3, s16 | ND | - | - | 2.2% | 0.9% | 14% | 97% | 25% | 86% |
| IKBKB | PHD1 | Trypsin | ELDQGSLC(+57.02)TSFVGTLQYLAPELLEQQK | 3, s17 | ND | - | - | 14% | 14% | 14% | 97% | 25% | 86% |
| MAPK6 | PHD3 | Trypsin | YMDLKPLGC(+57.02)GGNGLVFSAVDNDC(+57.02)DKR | 3 | M21 | 19% | 12% | 0.3% | 0.3% | 9.5% | 15% | 23% | 94% |
| NDRG3 | PHD2 | Trypsin | MADC(+57.02)GGLPQVVQPGK | 3, s18 | M287 | 48% | 21% | 1.8% | 0.9% | 17% | 92% | 24% | 93% |
| PDE4D | PHD2 | Trypsin | LMHSSSLTNSSIPR | 3, s19 | M371 | 61% | 36% | ND | ND | 0.5% | 88% | 36% | 93% |
| | | | IAELSGNRPLTVIMHTIFQER | 3, s20 | M424 | 51% | 45% | ND | ND | | | | |

*Table 2 continued on next page*

*Table 2 continued*

| Substrate | PHD Isoform | Protease | Quantified peptide | Figure | Oxidation | Assigned ox (%) Control | Assigned ox (%) Enzyme | Unassigned ions (%) Control | Unassigned ions (%) Enzyme | HIF-1α P402ox (%) Control | HIF-1α P402ox (%) Enzyme | HIF-1α P564ox (%) Control | HIF-1α P564ox (%) Enzyme |
| --- | --- | --- | --- | --- | --- | --- | --- | --- | --- | --- | --- | --- | --- |
| | | | | | | | | | | | Non-HIF substrate | | HIF-1α control |
| PKM | PHD3 | Trypsin | LAPITSDPTEATAVGAVEASFK | 3, s21 | ND | - | - | <0.1% | 0.1% | 6.3% | 9.9% | 29% | 91% |
| PPP2R2A | PHD2 | Trypsin | IWDLNMENRPVETYQVHEYLR | 3, s22 | M315 | 14% | 20% | ND | ND | 27% | 86% | 36% | 91% |
| SPRY2 | PHD1 | Trypsin | AQSGNGSQPLLQTPR | - | - | - | - | ND | ND | - | - | - | - |
| | PHD3 | | | 3, s23 | ND | - | - | ND | ND | ND | ND | 52% | 92% |
| | PHD1 | | LLGSSFSSGPVADGIIR | 3, s24 | ND | - | - | 0.9% | ND | 9.5% | 92% | 23% | 83% |
| | PHD3 | | | 3, s25 | ND | - | - | ND | ND | ND | ND | 52% | 92% |
| | PHD1 | | SELKPGELKPLSK | 3, s26 | ND | - | - | ND | ND | 9.5% | 92% | 23% | 83% |
| | PHD3 | | | 3, s27 | ND | - | - | ND | ND | ND | ND | 52% | 92% |
| TELO2 | PHD3 | Trypsin | AVLIC(+57.02)LAQLGEPELR | 3, s28 | ND | - | - | 8.9% | 6.5% | ND | ND | 52% | 92% |
| THRA | PHD2 | Trypsin | SLQQRPEPTPEEWDLIHIATEAHR | 3, s29 | ox: NL | 0.9% | 0.7% | 1.8% | 1.3% | 15% | 94% | 26% | 96% |
| | | | | | diox: W165 | 5.4% | 5.1% | | | | | | |
| | PHD3 | | | 3, s30 | ox: NL | 0.9% | 0.8% | 1.8% | 0.8% | 15% | 16% | 26% | 86% |
| | | | | | diox: W165 | 5.4% | 4.8% | | | | | | |
| TP53 | PHD1 | Trypsin | TC(+57.02)PVQLWVDSTPPPGTR | 3, s31 | ND | - | - | ND | ND | 9.5% | 92% | 23% | 83% |
| | PHD3 | LysC | EPGGSRAHSSHLK | 3, s32 | ND | - | - | ND | 0.4% | 21% | 24% | 30% | 78% |
| TRPA1 | PHD2 | Trypsin | NLRPEFMQMQQIK | 3, s33 | M397 | 31% | 30% | ND | ND | 0.5% | 88% | 36% | 93% |
| | | | | | M399 | 33% | 34% | | | | | | |

DOI: https://doi.org/10.7554/eLife.46490.007

The following source data is available for Table 2:

**Source data 1.** Peptide standards employed in IVTT hydroxylation assays.

The table lists synthetic peptide sequences corresponding to unoxidised and hydroxylated variants (column 3) of protease-digested peptides assigned and quantified in IVTT hydroxylation assays. Equimolar injections of the indicated peptide variants were used for comparison of detection efficiency (column 4) and chromatographic elution time (column 5) by LC-MSMS analysis. References to the primary XIC data are indicated in column 6.

DOI: https://doi.org/10.7554/eLife.46490.008

MS analyses were initially performed using an ESI-LC-MS platform (QExactive, Thermo Scientific) in data-dependent acquisition (DDA) mode. Using this approach, we were able to detect substrate peptides containing the target prolyl residue(s) in non-hydroxylated form with high confidence. One drawback of DDA, however, is that only a subset of the most abundant precursor ions are isolated and fragmented by MSMS. The stochastic nature of precursor ion selection raised the possibility that low stoichiometry hydroxylation(s) of lesser abundance might not be selected for fragmentation. To address this limitation, we also measured the abundance of precursor ions in the raw data. Guided by previous work in which hydroxylation had a relatively modest effect on chromatographic behaviour (*Singleton et al., 2014*; *Singleton et al., 2011*), we considered precursor ions that eluted up to 5 min in advance of the non-hydroxylated ion under reverse-phase chromatography conditions. This window was subsequently guided by the elution characteristics of peptide standards.

The results are summarised in *Table 2*. MSMS assigned potential oxidations (+16 Da shifts) are indicated and displayed alongside manually-curated precursor ion quantitation, in which the oxidation is expressed as a percentage relative to the unmodified peptide. For each batch of analyses, quantification of hydroxylation on the control HIF-1$\alpha$ polypeptide was carried out. Since the lysates used in IVTT reactants contain endogenous PHD activity, significant levels of HIF-1$\alpha$ hydroxylation were observed in IVTTs without addition of recombinant PHDs. However, robust increases in HIF-1$\alpha$ hydroxylation were always observed following addition of a recombinant PHD, and are reported in columns 11–14 of *Table 2*; note that PHD1 and PHD2 catalysed hydroxylations at both P402 and P564, whereas PHD3 had little activity on P402, consistent with known HIF substrate specificity (*Hirsilä et al., 2003*). To provide a further positive control we also tested whether prolyl hydroxylation at the reported sites in all three HIF-$\alpha$ isoforms was supported in these assays. Therefore, in a further series of experiments, IVTTs of HIF-1$\alpha$, HIF-2$\alpha$ and HIF-3$\alpha$ were reacted in parallel with each of the PHDs. These experiments confirmed robust prolyl hydroxylation at P402 and P564 in HIF-1$\alpha$, P405 and P531 in HIF-2$\alpha$ and P492 in HIF-3$\alpha$, with both PHD1 and PHD2 and robust hydroxylation of HIF-1$\alpha$ P564, HIF-2$\alpha$ P531 and HIF-3$\alpha$ P492, but not at the other two sites for PHD3, consistent with the known site specificity of PHD3 (*Figure 2* and supplements).

By contrast, prolyl hydroxylation was not assigned on any of the reported non-HIF substrate sites by the software search, even though the non-oxidised target peptide was always identified. A number of peptides spanning the reported site of prolyl hydroxylation bore putative hydroxylations (i. e. +16 Da mass increments) that could be assigned to residues other than proline. These were most frequently on methionine, but also included aspartate and di-hydroxylation of tryptophan; in three substrates the putative oxidation event could not be localised to a specific residue. These assignments are indicated in *Table 2*, column 6.

Automated analyses were followed by in-depth manual analyses. Extracted ion chromatograms (XICs) corresponding to the mass-to-charge (m/z) ratio of assigned unmodified precursor ions and their observed or putative oxidised species (as determined by mass shift) were compared between control and PHD supplemented reactions, giving a complete view of all relevant precursor ions within the stated retention time window. Panel B of *Figure 3* illustrates an example of such an analysis on the reported substrate mitogen-activated protein kinase 6; panel B of *Figure 3—figure supplements 1–33* to *Figure 3* illustrate these analyses for each putative substrate prepared by IVTT and reacted with the relevant PHD enzyme. Peak area integrations were performed on all MSMS assigned species and compared between PHD-reacted and control IVTTs; this XIC-derived quantitation is summarised in columns 7 and 8, *Table 2*. These analyses also revealed unassigned ions of compatible m/z for oxidation that were typically of low stoichiometry relative to the non-hydroxylated species. These species were also quantified in PHD-reacted and control IVTTs, using similar methods; the data is summarised in columns 9 and 10 of *Table 2*. Neither the assigned oxidations, nor these putative oxidised species were significantly increased by reaction with the relevant PHD. Overall therefore, this analysis provided no evidence for prolyl hydroxylation at the proposed sites on the non-HIF target polypeptides, and no evidence for PHD enzyme-dependent oxidation of any type within the relevant tryptic digest peptides.

Nevertheless, given previous reports of PHD-catalysed prolyl hydroxylation, we considered the possibility that PHD-catalysed prolyl hydroxylations might have been missed through altered ionisation/detection efficiencies in our experiments, or might have been obscured by other putative oxidations (+16 Da shifts) on the peptides. Although these oxidations appeared to arise independently of

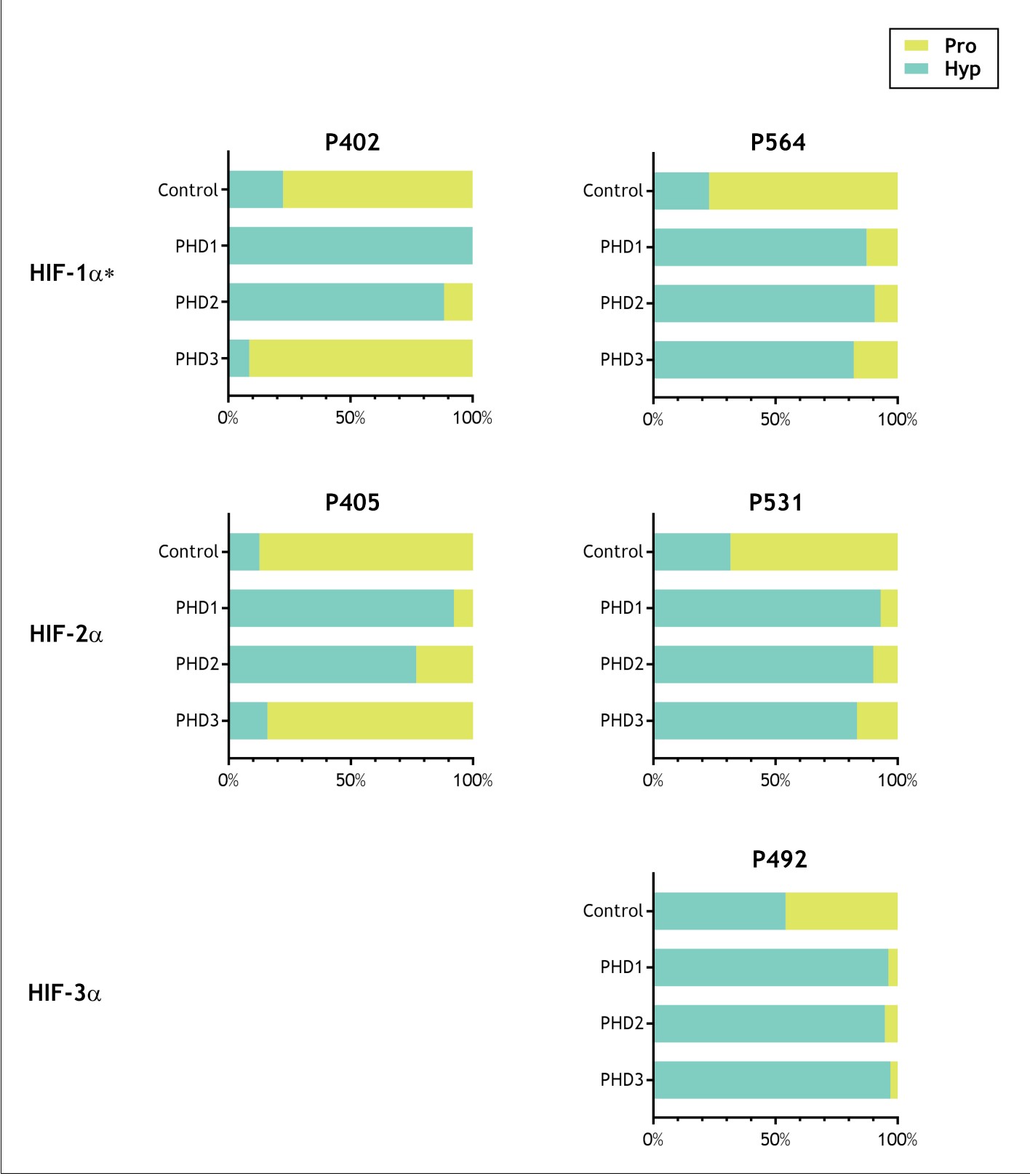

**Figure 2.** Activity of PHD enzymes on full-length HIF substrates produced by IVTT. HIF substrates (HIF-1α, HIF-2α and HIF-3α) were produced by IVTT and reacted in the absence (Control) or presence of the indicated recombinant PHD enzyme. Endogenous PHD activity in IVTT lysate gave rise to a basal (Control) level of hydroxylation that markedly increased upon addition of recombinant PHD. Data are summarised as stacked bar charts that are grouped by target site and report the (%) level of prolyl hydroxylation determined by LC-MSMS. Key: Pro (unoxidised prolyl, yellow), Hyp (hydroxyprolyl,

*Figure 2 continued on next page*

*Figure 2 continued*

turquoise). Extracted ion chromatograms corresponding to each hydroxylation reaction are provided as the following supplements to *Figure 2—figure supplement 1*, HIF-1α P402; *Figure 2—figure supplement 2*, HIF-1α P564; *Figure 2—figure supplement 3*, HIF-2α P405; *Figure 2—figure supplement 4*, HIF-2α P531; *Figure 2—figure supplement 5*, HIF-3α P492. *Modified HIF-1α sequence (M561A, M568A) assayed.
DOI: https://doi.org/10.7554/eLife.46490.009

The following figure supplements are available for figure 2:

**Figure supplement 1.** Quantitation of PHD-catalysed hydroxylation at the P402 site in hypoxia-inducible factor 1α.
DOI: https://doi.org/10.7554/eLife.46490.010

**Figure supplement 2.** Quantitation of PHD-catalysed hydroxylation at the P564 site in hypoxia-inducible factor 1α.
DOI: https://doi.org/10.7554/eLife.46490.011

**Figure supplement 3.** Quantitation of PHD-catalysed hydroxylation at the P405 site in hypoxia-inducible factor 2α.
DOI: https://doi.org/10.7554/eLife.46490.012

**Figure supplement 4.** Quantitation of PHD-catalysed hydroxylation at the P531 site in hypoxia-inducible factor 2α.
DOI: https://doi.org/10.7554/eLife.46490.013

**Figure supplement 5.** Quantitation of PHD-catalysed hydroxylation at the P492 site in hypoxia-inducible factor 3α.
DOI: https://doi.org/10.7554/eLife.46490.014

PHD activity, it is possible that co-elution of a prolyl hydroxylated species with a non-enzymatically oxidised ion could confound the quantification of PHD-dependent hydroxylation, that is a high level of non-enzymatic oxidation could obscure a low level of PHD-dependent prolyl hydroxylation. We therefore synthesised each peptide of interest (i.e., the trypsin, or other protease, generated fragment spanning the reported target prolyl residue) in unhydroxylated (prolyl) and *trans* C-4 hydroxylated (4(*R*)-hydroxyprolyl) forms. These peptide standards (*Table 2—source data 1.*) were first used to determine whether prolyl hydroxylation resulted in major changes in ionisation efficiency for any of the species. To account for any differences in solubility or purity, weighed aliquots of peptides were first quantified by comparison with an internal trimethylsilylpropanoic acid (TSP) standard using $^{1}$H NMR spectroscopy. Equimolar quantities of prolyl- and hydroxyprolyl-peptides, as defined by NMR, were then analysed by MS on the same instrumentation, using identical acquisition conditions to those used for the IVTT hydroxylation assays. Accordingly, peptide intensities were quantified by peak area integration of extracted ion chromatograms and expressed as relative ratios. Related peptide sets displayed broadly comparable ion counts, as summarised in *Table 2—source data 1*. Although deviation from a 1:1 ratio was observed for certain pairs, this data does not support the possibility that failure to detect hydroxyprolyl peptides derived from IVTT products might be due to suppression of peptide ionisation.

The use of peptide standards also provided precise data on the effect of prolyl hydroxylation on chromatographic elution. A reproducible shift in chromatographic retention between related pairs of prolyl and hydroxyprolyl peptides was observed. This shift was variable across peptide pairs, ranging from −0.14 to −4.72 min for a given hydroxyprolyl modification resolved on a 60 min linear gradient (*Table 2—source data 1.*). This indicated that manual searches for oxidised peptides within a retention time window of −5 min to 0 min of the unoxidised peptide had covered all potential prolyl hydroxylated ions. Re-examination of the data, together with the performance of additional IVTT reactions and LC-MS runs in which peptide standards were run immediately after the IVTT samples, indicated that in the large majority (all except two), any oxidised ions were chromatographically distinct from the putative prolyl hydroxylated species (see panel A of *Figure 3* and *Figure 3—figure supplements 1–33*). This procedure provides security that prolyl hydroxylated species are unlikely to have been obscured.

To further address the assignment of potentially oxidised species on the two peptides, where we considered that there was a possibility that co-elution might impair detection of low levels of prolyl hydroxylation, we performed additional reactions and analysed the products by parallel reaction monitoring using an ESI-LC-MS spectrometer (Fusion Lumos, Thermo Scientific) in order to measure diagnostic fragment ions that are specific to the hydroxylation site on the precursor peptide. This procedure enabled the dependence of any potentially oxidised species on reaction with PHD to be defined by reference to specific fragment ions, in addition to extracted ion chromatograms. These analyses are illustrated in *Figure 4*. Despite robust detection of diagnostic fragment ions derived from hydroxyprolyl peptide standards, no such ions were observed in the IVTT material either before

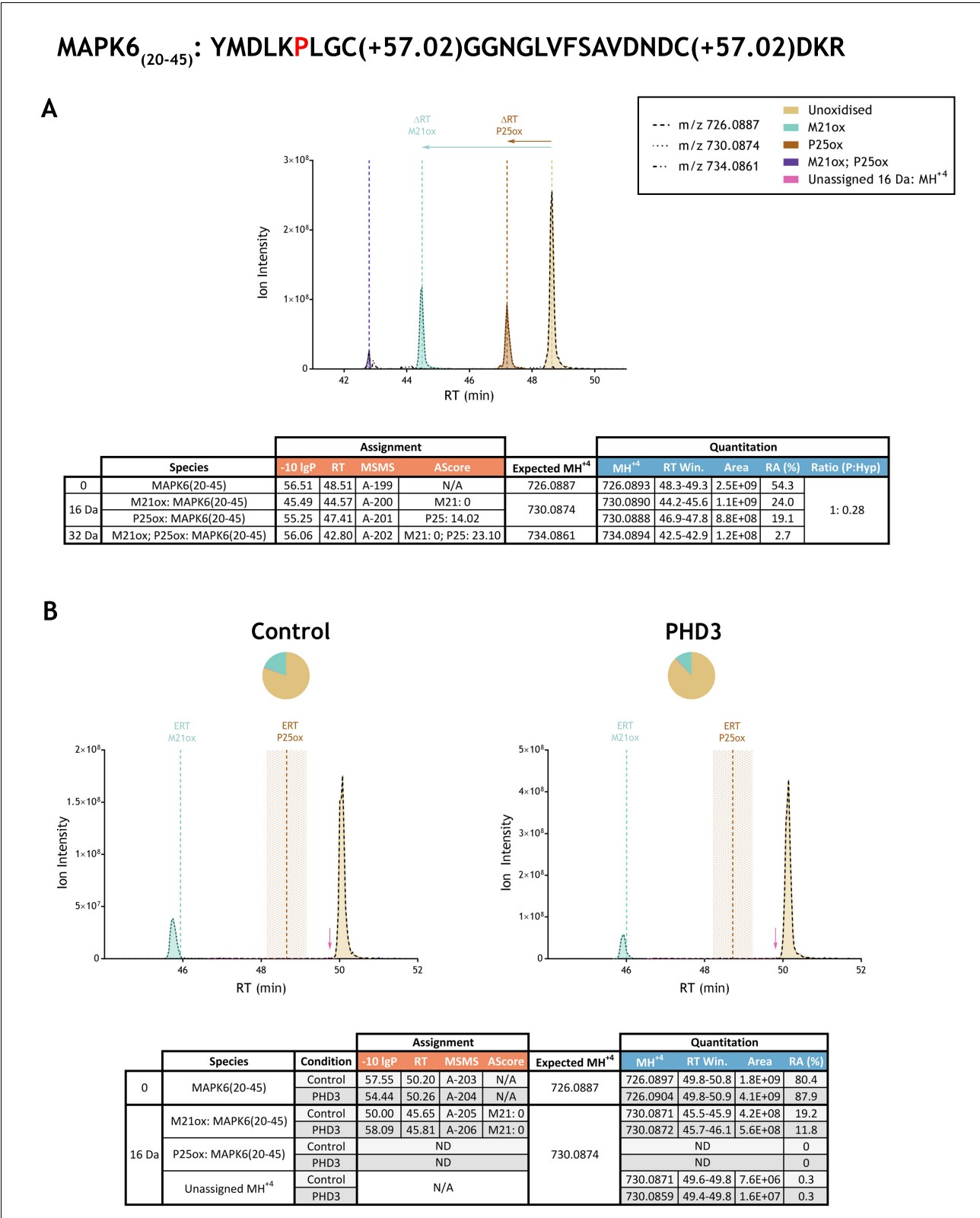

**Figure 3.** Example of quantitation of peptide oxidation on IVTT substrates reacted with PHD enzyme by extracted ion chromatogram (XIC). (**A**) Shows ion intensity and retention time (RT) characteristics of equimolar synthetic peptide standards for unoxidised and P25ox forms of the tryptic MAPK6(20–45) peptide. MSMS assigned species including non-enzymatic oxidations are coloured (see inset); vertical dashed lines define peak maxima used to derive ΔRT values for key oxidised ions. (**B**) Shows comparable XIC data for protease-digested IVTT substrates under control (left) or PHD3-reacted

*Figure 3 continued on next page*

*Figure 3 continued*

(right) conditions. Peaks corresponding to unoxidised and M21ox forms of MAPK6(20–45) were assigned by MSMS (see inset for colour code); P25ox was not detected. Estimated RT (ERT) values for oxidised peptides, derived from (**A**), are indicated by dashed vertical lines, shading applied to ERT P25ox corresponds to a 1 min RT window. Peptide ions of low abundance that were compatible with oxidation but not assigned by MSMS are also indicated (pink arrow). Quantitative data for observed species are presented as pie charts with XIC data. Assignment and quantitative data is provided below each panel as follows: (−10lgP) significance score of leading assignment at given (RT) with ambiguity score (AScore) for PTM localisation and reference to primary MSMS data in *Supplementary file 1*. Quantitative data reports ion counts (Area) of observed masses (MH$^{+4}$) integrated over time (RT Win.), expressed as relative abundance (RA). Data on detection efficiency for synthetic peptides (Panel A, P: Hyp) is an aggregated ratio of prolyl (P) to hydroxyprolyl (Hyp) species.

DOI: https://doi.org/10.7554/eLife.46490.015

The following figure supplements are available for figure 3:

**Figure supplement 1.** Quantitation of peptide oxidation on ACACB(341-366) following reaction of full-length acetyl-CoA carboxylase 2 with recombinant PHD3.

DOI: https://doi.org/10.7554/eLife.46490.016

**Figure supplement 2.** Quantitation of peptide oxidation on ACACB(436-454) following reaction of full-length acetyl-CoA carboxylase 2 with recombinant PHD3.

DOI: https://doi.org/10.7554/eLife.46490.017

**Figure supplement 3.** Quantitation of peptide oxidation on ACTB(292-312) following reaction of full-length beta-actin with recombinant PHD3.

DOI: https://doi.org/10.7554/eLife.46490.018

**Figure supplement 4.** Quantitation of peptide oxidation on ACTB(316-326) following reaction of full-length beta-actin with recombinant PHD3.

DOI: https://doi.org/10.7554/eLife.46490.019

**Figure supplement 5.** Quantitation of peptide oxidation on ADRB2(376–404) following reaction of full-length beta-2 adrenergic receptor with recombinant PHD3.

DOI: https://doi.org/10.7554/eLife.46490.020

**Figure supplement 6.** Quantitation of peptide oxidation on AKT1(122–140) following reaction of full-length AKT serine/threonine kinase 1 with recombinant PHD2.

DOI: https://doi.org/10.7554/eLife.46490.021

**Figure supplement 7.** Quantitation of peptide oxidation on AKT1(308–328) following reaction of full-length AKT serine/threonine kinase 1 with recombinant PHD2.

DOI: https://doi.org/10.7554/eLife.46490.022

**Figure supplement 8.** Quantitation of peptide oxidation on AKT1(421–436) following reaction of full-length AKT serine/threonine kinase 1 with recombinant PHD2.

DOI: https://doi.org/10.7554/eLife.46490.023

**Figure supplement 9.** Quantitation of peptide oxidation on ATF4(142–168) following reaction of full-length activating transcription factor 4 with recombinant PHD3.

DOI: https://doi.org/10.7554/eLife.46490.024

**Figure supplement 10.** Quantitation of peptide oxidation on ATF4(172–198) following reaction of full-length activating transcription factor 4 with recombinant PHD3.

DOI: https://doi.org/10.7554/eLife.46490.025

**Figure supplement 11.** Quantitation of peptide oxidation on CENPN(308-329) following reaction of full-length centromere protein N with recombinant PHD2.

DOI: https://doi.org/10.7554/eLife.46490.026

**Figure supplement 12.** Quantitation of peptide oxidation on CEP192(2306–2317) following reaction of full-length centrosomal protein 192 (isoform 1) with recombinant PHD1.

DOI: https://doi.org/10.7554/eLife.46490.027

**Figure supplement 13.** Quantitation of peptide oxidation on EEF2K(94-111) following reaction of full-length eukaryotic elongation factor 2 kinase with recombinant PHD2.

DOI: https://doi.org/10.7554/eLife.46490.028

**Figure supplement 14.** Quantitation of peptide oxidation on EPOR(426-453) following reaction of the cytoplasmic domain of erythropoietin receptor (274-508) with recombinant PHD3.

DOI: https://doi.org/10.7554/eLife.46490.029

**Figure supplement 15.** Quantitation of peptide oxidation on FLNA(2311–2333) following reaction of full-length filamin A with recombinant PHD2.

DOI: https://doi.org/10.7554/eLife.46490.030

**Figure supplement 16.** Quantitation of peptide oxidation on FOXO3(420–444) following reaction of full-length forkhead box O3 with recombinant PHD1.

DOI: https://doi.org/10.7554/eLife.46490.031

*Figure 3 continued on next page*

*Figure 3 continued*

**Figure supplement 17.** Quantitation of peptide oxidation on IKBKB(172-198) following reaction of full-length inhibitor of nuclear factor kappa B kinase subunit beta with recombinant PHD1.
DOI: https://doi.org/10.7554/eLife.46490.032

**Figure supplement 18.** Quantitation of peptide oxidation on NDRG3(287–301) following reaction of *N*-terminally truncated NDRG3 (residues 108–375) with recombinant PHD2.
DOI: https://doi.org/10.7554/eLife.46490.033

**Figure supplement 19.** Quantitation of peptide oxidation on PDE4D(370-383) following reaction of full-length phosphodiesterase 4D (isoform 6) with recombinant PHD2.
DOI: https://doi.org/10.7554/eLife.46490.034

**Figure supplement 20.** Quantitation of peptide oxidation on PDE4D(411-431) following reaction of full-length phosphodiesterase 4D (isoform 6) with recombinant PHD2.
DOI: https://doi.org/10.7554/eLife.46490.035

**Figure supplement 21.** Quantitation of peptide oxidation on PKM(401-422) following reaction of full-length pyruvate kinase M2 with recombinant PHD3.
DOI: https://doi.org/10.7554/eLife.46490.036

**Figure supplement 22.** Quantitation of peptide oxidation on PPP2R2A(310-330) following reaction of full-length protein phosphatase 2 regulatory subunit B α with recombinant PHD2.
DOI: https://doi.org/10.7554/eLife.46490.037

**Figure supplement 23.** Quantitation of peptide oxidation on SPRY2(5–19) following reaction of full-length sprouty homolog 2 with recombinant PHD3.
DOI: https://doi.org/10.7554/eLife.46490.038

**Figure supplement 24.** Quantitation of peptide oxidation on SPRY2(135–151) following reaction of full-length sprouty homolog 2 with recombinant PHD1.
DOI: https://doi.org/10.7554/eLife.46490.039

**Figure supplement 25.** Quantitation of peptide oxidation on SPRY2(135–151) following reaction of full-length sprouty homolog 2 with recombinant PHD3.
DOI: https://doi.org/10.7554/eLife.46490.040

**Figure supplement 26.** Quantitation of peptide oxidation on SPRY2(156–168) following reaction of full-length sprouty homolog 2 with recombinant PHD1.
DOI: https://doi.org/10.7554/eLife.46490.041

**Figure supplement 27.** Quantitation of peptide oxidation on SPRY2(156–168) following reaction of full-length sprouty homolog 2 with recombinant PHD3.
DOI: https://doi.org/10.7554/eLife.46490.042

**Figure supplement 28.** Quantitation of peptide oxidation on TELO2(363–377) following reaction of full-length telomere maintenance 2 with recombinant PHD3.
DOI: https://doi.org/10.7554/eLife.46490.043

**Figure supplement 29.** Quantitation of peptide oxidation on THRA(153-176) following reaction of full-length thyroid hormone receptor alpha with recombinant PHD2.
DOI: https://doi.org/10.7554/eLife.46490.044

**Figure supplement 30.** Quantitation of peptide oxidation on THRA(153-176) following reaction of full-length thyroid hormone receptor alpha with recombinant PHD3.
DOI: https://doi.org/10.7554/eLife.46490.045

**Figure supplement 31.** Quantitation of peptide oxidation on TP53(140-156) following reaction of full-length tumour protein p53 with recombinant PHD1.
DOI: https://doi.org/10.7554/eLife.46490.046

**Figure supplement 32.** Quantitation of peptide oxidation on TP53(358-370) following reaction of full-length tumour protein p53 with recombinant PHD3.
DOI: https://doi.org/10.7554/eLife.46490.047

**Figure supplement 33.** Quantitation of peptide oxidation on TRPA1(391–403) following reaction of full-length transient receptor potential cation channel subfamily A member 1 with recombinant PHD2.
DOI: https://doi.org/10.7554/eLife.46490.048

or after reaction with the relevant PHD enzyme. Thus MS-based analyses of reported non-HIF PHD substrates did not support the catalysis of detectable levels of hydroxylation, at least under these conditions.

To further consider the possibility that these assays might not have detected hydroxylation of prolyl residues, we conducted direct assays for hydroxyproline using a radiochemical assay. For

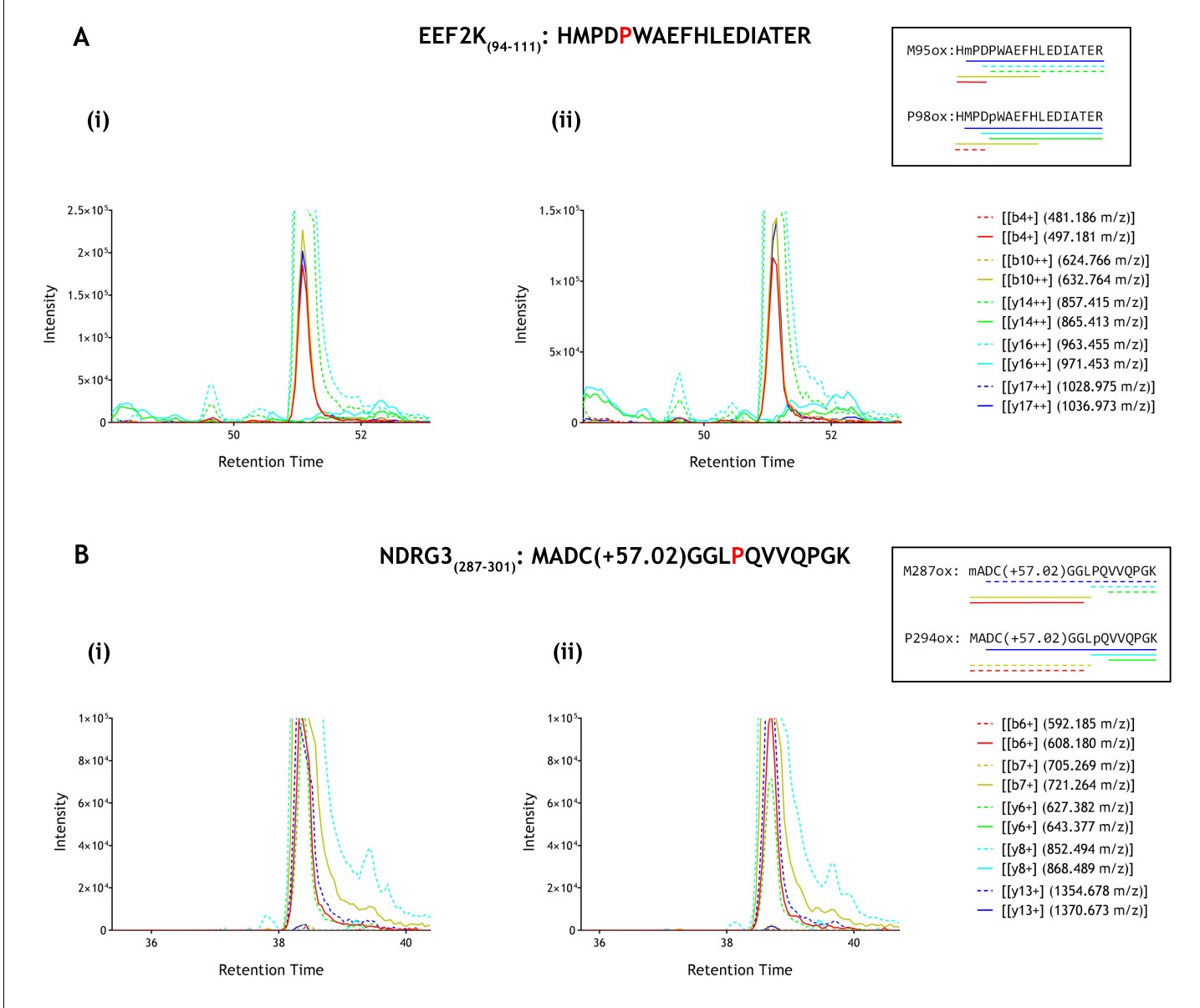

**Figure 4.** Parallel reaction monitoring (PRM) of oxidised EEF2K(94-111) and NDRG3(287–301) peptide species by mass spectrometry. Fragment ions (n = 5), including ions of discriminatory mass for proline and methionine oxidation (see inset for schematic representation of fragment ions; dashed lines: unoxidised, solid lines: oxidised) were selected for PRM acquisition, based on existing MSMS data (*Supplementary file 1*: A167-169 and A207-209). The figure shows XIC data of PRM fragment ions corresponding to oxidised forms of (**A**) EEF2K(94-111) and (**B**) NDRG3(287–301) derived from IVTT hydroxylation assays under (**i**) control or (**ii**) PHD2-reacted conditions. Fragment ion masses indicative of proline oxidation were not observed across the elution profile. Note, the y-axis has been truncated to show fragment ions of lesser abundance (peak maxima for panels A and B: >1×10$^6$ ion counts).

DOI: https://doi.org/10.7554/eLife.46490.049

these assays, the IVTTs representing the reported polypeptide were produced and reacted with the relevant PHD using the same procedures as for the MS analysis, except for the use of radiolabelled L-[2,3,4,5-$^3$H]proline in the IVTT reaction (*Koivunen et al., 2006*). Following the substrate-PHD reaction, the samples were dialysed extensively to remove any remaining free $^3$H-proline, and the 4-hydroxy-$^3$H-proline formed in the substrate was analysed by a specific radiochemical procedure (*Juva and Prockop, 1966*). The results are provided in *Figure 5*. Despite the presence of only two

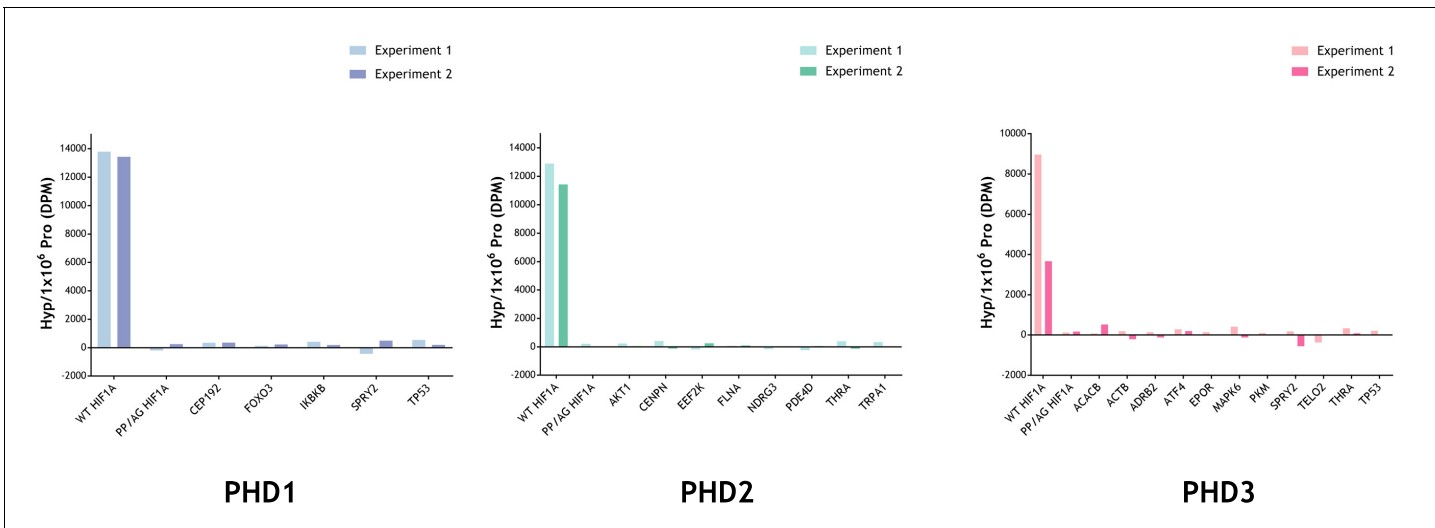

**Figure 5.** Radiochemical assay measuring the relative amount of 4-Hydroxy[3H]proline formed after incubation of reported substrate with PHD enzyme. Substrates were produced by IVTT in the presence of *L*-[2,3,4,5-3H]proline and incubated with and without the indicated recombinant PHD enzyme. Conversion to 4-hydroxy[3H]proline was measured by radiochemical assay with data expressed as DPM Hyp/1 × 10[6] DPM Pro, DPM of the reaction without the PHD being subtracted. Assay efficacy was confirmed with a positive HIF-1α (WT) control. Background DPM range was determined with a negative PP/AG HIF-1α (P402A, P564G proline mutant) control. The hydroxylation level observed in PHD-reacted non-HIF substrates was not above background. Data are from two independent assays with the following (n = 1) exceptions: PKM (PHD3); TELO2 (PHD3); TRPA1 (PHD2).
DOI: https://doi.org/10.7554/eLife.46490.050

The following source data is available for figure 5:

**Source data 1.** Numerical data for 4-hydroxy[3H]proline assay represented in *Figure 5*.
DOI: https://doi.org/10.7554/eLife.46490.051

target prolyl residues in HIF-1α, robust PHD-dependent hydroxylation was detected. However, no PHD-dependent hydroxylation was detected on any of the other polypeptides.

## Discussion

Multiple potential non-HIF substrates of the PHD enzymes have been reported using a range of methods, including interaction assays with PHDs or pVHL, analyses of proteomic responses to hypoxia or 2-oxoglutarate dioxygenase inhibitors (e.g. dimethyloxalylglycine), or through candidacy inferred from cell biological experiments. The presence of hydroxyproline was usually assigned either by MSMS or by use of anti-hydroxyprolyl antibodies, though in some cases the modification was not demonstrated directly. In some cases, in vitro reactions using isolated PHDs were performed to support the proposed assignment of PHD-catalysed prolyl hydroxylation. These approaches present challenges *en route* to the confident assignment of post-translational modifications; in particular, in the specificity of immunodetection using antibodies, in the accurate assignment of peptide oxidations, and in distinguishing between enzymatic and artefactual oxidation. Nevertheless, we were surprised by the discrepancy between the reported assignments of prolyl hydroxylation and the results of our study.

Using several different MS methods, including detailed manual analysis of the data guided by unmodified and hydroxyprolyl-modified peptide standards, we were unable to detect catalytic activity of the PHDs on any of the reported non-HIF substrates. We were also unable to detect such activity using radiochemical assays monitoring conversion of proline to hydroxyproline in the full-length target proteins.

In the course of the work we considered several reasons for this discrepancy. Firstly, it could be that, despite evidence that HIF-α peptides initially bind to PHD2 in a substantially unstructured conformation (*Chowdhury et al., 2009*), longer polypeptides might be necessary for this interaction with non-HIF substrates, as is the case for some other 2OG dependent protein hydroxylases (*Feng et al., 2014*). We therefore performed extensive additional experiments using full-length

polypeptide substrates. Secondly, it could be that despite high activity of the isolated catalytic domains of the PHDs on HIF-α, full-length enzymes may be necessary for activity on non-HIF substrates. We therefore performed assays using preparations of the full-length enzyme. Thirdly, we were concerned that our MS analyses might have overlooked hydroxylated species. To minimise this possibility, we used peptide standards to test whether *trans* C-4 prolyl-hydroxylation affected ionisation efficiency and to enable distinction from confounding oxidised species. We also carried out radiochemical assays for hydroxylation monitoring conversion of [$^3$H]-proline to 4-hydroxy-[$^3$H]-proline. As with the peptide work, these experiments included attempts to enhance the hydroxylation of non-HIF substrates using several times the quantity of PHD enzyme that was necessary to drive the hydroxylation of HIF substrates to near completion. None of these series of experiments revealed any evidence for catalytic hydroxylation activity on any of the reported non-HIF substrates.

The reasons for the discrepancy between our results and the published reports of non-HIF PHD substrates that prompted this survey are unclear. It is possible that the *bona fide* hydroxylation rates with isolated enzymes/substrates are much less than those for HIF-α peptides and were below the detection limits of our assays. It is also possible that PHD-catalysed hydroxylation might take place in cells but not in our assays. For instance, some or all reported non-HIF substrates might require one or more adaptor molecules, which are unnecessary for HIF substrates, to promote hydroxylation.

Several large 'in depth' MS based analyses of cellular proteomes are reported (*Bekker-Jensen et al., 2017*; *Geiger et al., 2012*; *Davis et al., 2017*). Although cells and culture conditions used in these experiments were not necessarily identical to those in the published reports of PHD-catalysed prolyl hydroxylation, this *in cellulo* data can be interrogated for evidence of prolyl hydroxylation on the proposed non-HIF substrates. We therefore analysed the deposited data using the same processing pipeline as that used in our analyses of the PHD-reacted polypeptides produced by IVTT. The results are summarised in *Table 3*. Among 12.8 million assigned spectra, we observed a total of 18,192 that corresponded to the trypsinolysis-derived peptides containing the proposed site (s) of prolyl hydroxylation in 17 of the non-HIF PHD substrates; in the remaining six substrates the target peptide was not identified. A total of 5038 putative singly oxidised peptides were assigned among 13 of these 17 identified sequences. Only 67 of these spectra were assigned as potential prolyl hydroxylations. However, inspection of the data revealed that all the computationally assigned putative prolyl oxidations were either at low fragment ion intensity and/or of low confidence with respect to the localisation of the oxidised residue. Most of these computer assignments of prolyl hydroxylation were on β-actin (ACTB) proline 322. These observations led us to consider the possibility that there are specific peptides that are prone to non-enzymatic oxidation and for which, because of the sequence context, PTM localisation is difficult. To this end, we applied a similar processing pipeline to spectra generated by the analyses of the synthetic unmodified (i.e. unhydroxylated) peptides used as standards for our earlier analyses of IVTT produced polypeptides. These analyses revealed a high prevalence of oxidation, some of which was computer assigned as hydroxyproline, on several of the peptides, including ACTB residues 316–326, AKT1 residues 308–328, and MAPK6 residues 20–45 (*Table 3—source data 1.*).

Putative oxidations were also observed on residues close to the target prolyl residue in the in vitro assays and in some cases, low abundance oxidations were observed that could not be assigned robustly from simple analysis of the fragmentation pattern. The use of peptide standards provided a means to assign these species, or at least distinguish them from hydroxyprolyl containing species and accurately assess the dependence of the modification on PHD enzyme activity. Although shifts in chromatographic retention were reproducible for a given pair of prolyl- and hydroxyprolyl- bearing peptides, there were large differences in this shift across different peptide pairs. This means that chromatographic differences between prolyl-hydroxylated and other oxidised species cannot be predicted, increasing the risks of confusion if standards are not used.

Difficulties in confirming reported substrate repertoires have been highlighted for other enzymes, leading to guidelines on the need for more complete biochemical analysis of the enzyme dependence of the proposed protein modification (*Kudithipudi and Jeltsch, 2016*). Given the potential for artefactual oxidations to confound MS analyses of prolyl hydroxylation, that was apparent in this and other studies (*Arsenault et al., 2015*), care is also required in assigning new substrates to the PHD enzymes. This includes the need for accurate location of the proposed prolyl oxidation event, and a clear demonstration of its dependence on PHD-

**Table 3.** Analysis of the cellular proteome for oxidised peptides encompassing reported sites of prolyl hydroxylation.

The number of unmodified and singly oxidised peptides containing reported sites of PHD-catalysed hydroxylation were counted over a range of stringency filters. Where multiple reported oxidation sites of one protein occur on separate peptides the number of spectra recorded is a summation of all interrogated peptides. The number of assigned target site prolyl oxidations is indicated in bold; methionine (Met) and alternate oxidation sites, including non-target proline residues (Other) are also presented together with unoxidised peptides for comparison. Stringency filters were applied as follows; PTM assignment confidence as assessed by ambiguity score (AScore:>20) and fragment ion intensity (Ion Intensity:>5%). These filters were applied separately and in combination to derive a list of assigned peptides with confidently localised oxidations. There are no high confidence proline oxidation assignments (i.e., meeting both AScore and Ion Intensity thresholds) of the reported substrates, suggesting a high degree of uncertainty from the software.

| Gene ID | Reported site | Assigned | # Spectra filtered by confidence of modification site | | | |
|---|---|---|---|---|---|---|
| | | | No filter | AScore | Ion Intensity | AScore and Ion Intensity |
| ACACB | P343, P450, P2131 | Unoxidised | 63 | N/A | N/A | N/A |
| ACTB | P307, P322 | **P322** | **57** | **7** | **11** | - |
| | | Met | 4528 | 647 | 4096 | 406 |
| | | Other | 336 | 58 | 62 | 7 |
| | | Unoxidised | 8650 | N/A | N/A | N/A |
| ADRB2 | P382*, P395* | Unoxidised | 1 | N/A | N/A | N/A |
| AKT1 | P125, P313*, P318*, P423 | Unoxidised | 160 | N/A | N/A | N/A |
| CENPN | P311 | Unoxidised | 7 | N/A | N/A | N/A |
| EEF2K | P98 | Met | 5 | - | - | - |
| | | Unoxidised | 41 | N/A | N/A | N/A |
| FLNA | P2317*, P2324* | **P2324** | **2** | - | - | - |
| | | Other | 1 | - | - | - |
| | | Unoxidised | 562 | N/A | N/A | N/A |
| FOXO3 | P426*, P437* | Unoxidised | 4 | N/A | N/A | N/A |
| NDRG3 | P294 | Met | 13 | 5 | - | - |
| | | Unoxidised | 49 | N/A | N/A | N/A |
| PDE4D | P29, P382, P419 | Met | 14 | 13 | 9 | 9 |
| | | Other | 1 | - | - | - |
| | | Unoxidised | 23 | N/A | N/A | N/A |
| PKM | P403*, P408* | **P403** | **3** | **1** | - | - |
| | | **P408** | **5** | - | - | - |
| | | Other | 5 | - | - | - |
| | | Unoxidised | 3394 | N/A | N/A | N/A |
| PPP2R2A | P319 | Met | 50 | 11 | 26 | 11 |
| | | Other | 18 | - | - | - |
| | | Unoxidised | 119 | N/A | N/A | N/A |
| SPRY2 | P18, P144, P160 | Unoxidised | 7 | N/A | N/A | N/A |
| TELO2 | P374, P419, P422 | Unoxidised | 47 | N/A | N/A | N/A |
| THRA | P160*, P162* | Unoxidised | 20 | N/A | N/A | N/A |
| TP53 | P142, P359 | Unoxidised | 7 | N/A | N/A | N/A |

*Table 3 continued on next page*

*Table 3 continued*

| Gene ID | Reported site | Assigned | # Spectra filtered by confidence of modification site | | | |
|---|---|---|---|---|---|---|
| | | | No filter | AScore | Ion Intensity | AScore and Ion Intensity |
| | Total | Pro | 67 | 8 | 11 | 0 |
| | | Met | 4610 | 676 | 4131 | 426 |
| | | Other | 361 | 58 | 62 | 7 |
| | | Unoxidised | 13154 | 0 | 0 | 0 |
| | | All | 18192 | 742 | 4204 | 433 |

*Doubly oxidised peptides were interrogated when multiple prolyl hydroxylation sites had been reported on the same tryptic peptide.
DOI: https://doi.org/10.7554/eLife.46490.052
The following source data is available for Table 3:
Source data 1. Summary of non-enzymatic oxidation assignments on synthetic peptide standards.
MSMS assignment frequency of artefactual oxidations observed on unmodified tryptic peptide standards; oxidations are stratified by residue (e.g., Met, Pro, other). Reported assignments (column 4) were not subject to PTM localisation (AScore) filtering. Oxidations assigned to target Pro residues are indicated in red. Variation in the total number of peptide assignments (Column 5) reflects differences in the amount of peptide injected and/or replicate runs.
DOI: https://doi.org/10.7554/eLife.46490.053
Source data 2. Peptide identification results from database search represented in *Table 3*.
DOI: https://doi.org/10.7554/eLife.46490.054

catalysis under conditions proposed to support hydroxylation. Our analyses clearly cannot disprove the presence of prolyl hydroxylation on reported non-HIF substrates under conditions other than those we have examined; they also do not exclude the possibility that the PHD enzymes have non-HIF substrates which have not yet been identified. However, they do suggest that the PHDs are relatively specific for their HIF-α substrates. This contrasts with the HIF asparaginyl hydroxylase, FIH (factor inhibiting HIF), which has been shown to catalyse hydroxylations at asparaginyl (and other) residues on a wide range of ankyrin repeat domain containing proteins, in addition to HIF-α (*Chowdhury et al., 2016*; *Elkins et al., 2003*). Consistent with this, FIH-catalysed asparaginyl hydroxylation of both HIF-α and non-HIF substrates was readily observed using the assays described in our current work (data not shown). These differences raise a question as to the underlying reasons for the apparent difference in selectivity between the two classes of HIF hydroxylase. X-ray and NMR structural analyses of HIF-α peptide in complex with the PHD catalytic domain reveal that the bound peptide makes multiple contacts with the enzyme (*Chowdhury et al., 2009*).There are also substantial conformational changes in the PHD structures on binding HIF-α substrates, including a loop which moves to tightly enclose the catalytic site and C-terminal region of the enzyme (*Chowdhury et al., 2016*). These observations contrast with FIH, which crystallographic studies imply has a much more accessible catalytic site and does not itself demonstrate a major conformational change upon substrate binding (*Elkins et al., 2003*).

These characteristics of PHDs likely contribute to their specificity for HIF-α substrates, but make it difficult predict other substrates from the primary sequence. Therefore, in an effort to better understand PHD substrate specificity, we compared both predicted and defined secondary structures across a range of HIF-α prolyl hydroxylation sites from different metazoan species and the reported non-HIF human substrates (*Table 1—source data 2.*). This revealed that despite considerable variation in primary sequence, most (possibly all) HIF-α peptides are predicted to form an α-helix to the N-terminal side of the target prolyl residue, as was observed in PHD2-HIF-α complex structures. In contrast, this feature was seen in few of the reported non-HIF substrates, where no clear pattern was evident. Although such an α-helix may therefore contribute to selectivity for HIF-α and/or guide future attempts to identify non-HIF substrates of the PHDs, a caveat is that the PHDs might have the capacity to generate major conformational changes in folded substrates, such as have been

described for the PHD homologue in *Pseudomonas aeruginosa* complexed with its EF-Tu (Elongation Factor Thermal unstable) substrate (*Scotti et al., 2014*).

# Materials and methods

## Key resources table

| Reagent type (species) or resource | Designation | Source or reference | Identifiers | Additional information |
|---|---|---|---|---|
| Gene (human) | ACACB | ORFeome collaboration | GenBank: BC172264 | |
| Gene (human) | ACTB | Mammalian Gene Collection | GenBank: BC001301.1 | |
| Gene (human) | ADRB2 | Mammalian Gene Collection | GenBank: BC073856.1 | |
| Gene (human) | AKT1 | ORFeome collaboration | GenBank: EU832571.1 | |
| Gene (human) | ATF4 | ORFeome collaboration | GenBank: DQ891758.2 | |
| Gene (human) | CENPN | Mammalian Gene Collection | GenBank: BC008972.1 | |
| Gene (human) | CEP192 | PMID: 23641073 | | |
| Gene (human) | EEF2K | ORFeome collaboration | GenBank: DQ894050.2 | |
| Gene (human) | EPOR | Sino Biological | NCBI Refseq: NR_033663.1 | |
| Gene (human) | FLNA | Addgene (#8982) | | |
| Gene (human) | FOXO3 | Mammalian Gene Collection | GenBank: BC020227.1 | |
| Gene (human) | IKBKB | Addgene (#11103) | | |
| Gene (human) | MAPK6 | ORFeome collaboration | GenBank: DQ894810.2 | |
| Gene (human) | NDRG3 | Mammalian Gene Collection | GenBank: BC144169.1 | |
| Gene (human) | PDE4D | ORFeome collaboration | GenBank: JF432192.1 | |
| Gene (human) | PKM | Mammalian Gene Collection | GenBank: BC012811.2 | |
| Gene (human) | PPP2R2A | Addgene (#13804) | | |
| Gene (human) | SPRY2 | ORFeome collaboration | GenBank: AM392994.1 | |
| Gene (human) | TELO2 | ORFeome collaboration | GenBank: AM392917.1 | |
| Gene (human) | THRA | ORFeome collaboration | GenBank: DQ895726.2 | |
| Gene (human) | TP53 | ORFeome collaboration | GenBank: DQ892492.2 | |
| Gene (human) | TRPA1 | ORFeome collaboration | GenBank: BC148423.1 | |
| Recombinant DNA reagent | 3xFLAG-tagged ORF | This study | | Vector backbone: pT7CFE1 (Thermo Fisher Scientific) |
| Recombinant protein | PHD3 (EGLN3) | PMID: 27502280 | | *E. coli* expression |

*Continued on next page*

*Continued*

| Reagent type (species) or resource | Designation | Source or reference | Identifiers | Additional information |
|---|---|---|---|---|
| Recombinant protein | PHD3 (EGLN3) | PMID: 12788921 | | Insect cell (Sf9) expression |
| Recombinant protein | PHD2 (EGLN1) | PMID: 12788921 | | Insect cell (Sf9) expression |
| Recombinant protein | PHD1 (EGLN2) | PMID: 12788921 | | Insect cell (Sf9) expression |
| Peptide | Assorted peptides | This study | | Sequence information provided in *Table 1—source data 1*. and *Table 2—source data 1*. |
| Commercial assay or kit | 1-Step Human Coupled IVT kit | Thermo Fisher Scientific | | |
| Software, algorithm | PEAKS Studio | Bioinformatics Solutions | | |

## Preparation of recombinant PHD enzymes in insect cells

Recombinant full-length human PHD1 and PHD2 with a *C*-terminal Flag-His6 tag and PHD3 with an *N*-terminal GST tag and a *C*-terminal Flag-His6 tag (*Hirsilä et al., 2005*; *Hirsilä et al., 2003*) were produced in H5 insect cells cultured in suspension in Sf900IISFM serum-free medium (Invitrogen). The cells were infected at a density of $1 \times 10^6$/ml with the recombinant baculoviruses at a multiplicity of 5 and harvested 72 hr after infection, washed with a solution of 150 mM NaCl and 20 mM phosphate, pH 7.4, and homogenised in a 0.1% Triton X-100, 150 mM NaCl, 100 mM glycine, 5 µM FeSO4, 10 µM DTT, 10 mM Tris buffer, pH 7.8 (4°C), supplemented with cOmplete EDTA-free protease inhibitor cocktail (Roche). The homogenate was centrifuged (10,000 rpm, 15 min) and the soluble fraction was filtered through a 0.45 µm filter and subjected to purification with an anti-FLAG M2 affinity gel (Sigma) equilibrated with 150 mM NaCl, 100 mM glycine, 10 µM DTT, 10 mM Tris buffer, pH 7.8 (4°C). A volume ratio of 1:5 of the affinity gel to soluble fraction of the insect cell homogenate was used and binding of the recombinant PHDs to the affinity gel was performed in a batch mode for 1 hr with gentle rotation at 4°C. The slurry was then poured into a chromatography column, the gel was allowed to settle, washing 4 times with 5 volumes of 150 mM NaCl, 5 µM FeSO4, 50 mM Tris buffer, pH 7.4 (4°C), supplemented with the EDTA-free protease inhibitor cocktail, and the recombinant PHDs were eluted with 150 µg/ml of FLAG peptide (Sigma) in the same buffer, elution volume being $3 \times 2$ ml per 1 ml of affinity gel.

## Preparation of recombinant PHD3 in *E. coli*

Full-length human PHD3 with a dual *N*-terminal thioredoxin-His6-tag was produced as described (*Chan et al., 2016*). *E. coli* BL21(DE3) cell cultures in 2TY medium were grown to OD600 of 0.6–0.8, then induced with isopropyl-β-D-thiogalactopyranoside (IPTG, 0.05 mM); growth was continued overnight at 18°C. Cells were lysed by sonication in Tris-HCl (20 mM, pH 7.5), NaCl (500 mM), imidazole (5 mM), glycerol (5%), DTT (5 mM), and PHD3 was purified by affinity chromatography using a His-trap column (GE Healthcare).

## Peptide synthesis

Peptides were either obtained commercially (ChinaPeptides or GLBiochem) or synthesised in-house on a Liberty Blue automated microwave peptide synthesiser (CEM). These peptides were prepared using Fluorenyl-methyloxycarbonyl (Fmoc) Rink amide MBHA resin (CS Bio), Nα-Fmoc-protected amino acids (Polymer labs), and the activators hydroxybenzotriazole (HOBt) and diisopropylcarbodiimide (DIC) at a ratio of 1:1 in DMF. Peptides were treated with a cleavage mix containing trifluoroacetic acid (TFA, 92.5% (v/v)), triisopropylsilane (2.5%), 1,3-dimethoxybenzene (2.5%) and water (2.5%). The cleaved peptides were precipitated by adding diethyl ether, lyophilised and purified by High Pressure Liquid Chromatography (HPLC).

HPLC-purified peptides employed as mass spectrometry standards in the IVTT hydroxylation assays (*Figure 3* and *Figure 3—figure supplements 1–33*) were synthesised in-house (Peptide

Chemistry STP, Francis Crick Institute) using Fmoc-protected amino acids on either Wang or Rink amide resin.

## Calibration of peptide standards

NMR peptide calibration experiments were performed using a Bruker AVIII 700 MHz spectrometer using an inverse TCI probe and 5 mm NMR tubes. $^1$H spectra were recorded with 296 transients. Peptide standards were prepared at an estimated concentration of 50 μM by weighing the lyophilised material and dissolving in 500 μL $D_2O$, supplemented with 1 μL of 10 mg/mL trimethylsilylpropanoic acid (TSP), an internal standard. The NMR-measured peptide concentrations were calculated by integration of the signal of the corresponding amino acid residues (in the 0–0.5 or 6–8 ppm region) compared to that of the internal standard at 0 ppm.

## Peptide hydroxylation assays

Electrospray-ionisation liquid chromatography–mass spectrometry (ESI-LC-MS) assays were performed using an ACQUITY Xevo G2-S QToF mass spectrometer (Waters). Recombinant enzyme (PHD1 and PHD2: 2 μM, PHD3: 10 μM) was incubated with peptide substrates (50 μM in assays using PHD1 and PHD2, 100 μM in assays using PHD3, sequences in *Table 1—source data 1.*) in the presence of (NH4)2Fe(II)(SO4)2 (50 μM), 2-oxoglutarate disodium salt (300 μM), and sodium L-ascorbate (300 μM) in Tris (50 mM, pH 7.5) for 4 hr at 37˚C. Reactions were quenched with formic acid (1 % v/v). The samples were separated on a ProSwift RP-1S Analytical column (Thermo Scientific) using a 10 min separation method with a gradient of solution A (Milli-Q H2O, 0.1% formic acid) and solution B (acetonitrile, 0.1% formic acid) under standard calibration. Instrument control and data analysis was conducted with MassLynx V4.1.

## Substrate production by IVTT

PHD protein substrates were prepared using an in vitro transcription and translation (IVTT) system derived from HeLa cell lysate (Thermo Scientific). Open reading frames (ORFs) were inserted into the pT7CFE1 expression vector (Thermo Scientific) encoding for proteins with a 3x FLAG epitope tag at either the *N*- or *C*-terminus of the protein. PCR and Gateway methods were employed, with the latter facilitated by the creation of a destination vector (pT7CFE 3xFLAG/Dest); sequence integrity was verified by Sanger sequencing. Substrate ORFs corresponded to canonical full-length human proteoforms (see *Table 1*) with the following exceptions referenced to Uniprot accession: CEP192 (Q8TEP8-1); EPOR (P19235-1; amino acids 274–508), NDRG3 (Q9UGV2-1; amino acids 108–375) PDE4D (Q08499-8). Substrates were translated for 3 hr at 30˚C, after which lysates were clarified by centrifugation (21,600 x *g* for 35 min) to remove insoluble aggregates (*Niwa et al., 2009*).

## IVTT hydroxylation assay and sample handling for MS

Hydroxylation assays were performed in batches with the amount of non-HIF substrate normalised to the HIF-1α comparator by FLAG immunoblotting. Substrate levels were also referenced to an internal bovine serum albumin standard to facilitate normalisation across batches. HIF-1α hydroxylation assays were performed in a reaction volume of 2.5 ml comprising 50 mM Tris pH 7.4, 28 nM 3x FLAG-HIF-1α (in HeLa IVTT lysate), 2 mg/ml BSA, 0.1 mM DTT, 0.06 mg/ml catalase, 2 mM ascorbate, 160 μM 2-oxoglutarate, 5 μM FeSO$_4$ and PHD enzyme (PHD1: 2.8 nM; PHD2: 0.56 nM; PHD3 14 nM) for 2 hr at 37˚C. For non-HIF substrates the amount of PHD enzyme was increased 2 to 5-fold. PHD-reacted substrates were immunopurified using FLAG magnetic beads (Sigma) and eluted with 100 μg/ml 3x FLAG peptide. Eluates were reduced and alkylated by treatment with DTT (5 mM; 1 hr at 25˚C) and iodoacetamide (20 mM; 1 hr at 25˚C) respectively, before methanol/chloroform precipitation/desalting. Samples were resuspended in urea buffer (6 M urea, 100 mM Tris pH 7.8) and sonicated to enhance solubilisation. Protease digestion used MS grade proteases (Trypsin from Sigma; Lys-C, from Promega; Asp-N and Glu-C from Roche; Elastase from Worthington Biochemical) and was performed under denaturing (1M urea) conditions. Peptides were purified and concentrated on C-18 spin columns (Thermo Scientific) prior to recovery in aqueous 2% (v/v) acetonitrile 0.1% (v/v) formic acid and subsequent mass spectrometric analysis. Peptide digests were multiplexed prior to mass spectrometric analysis as detailed in *Supplementary file 3*. The mass spectrometry proteomics (LC-MSMS) data have been deposited to the ProteomeXchange Consortium via the PRIDE partner

repository with the dataset identifier PXD013112 and 10.6019/PXD013112. Peptide LC-MS files have been archived at the Dryad digital repository and made publically available (DOI: 10.5061/dryad.917hb55).

## Mass spectrometry

Data dependent acquisitions (DDA) were acquired on a nano UPLC (Dionex Ultimate 3000; Thermo Scientific) coupled to a hybrid quadrupole Orbitrap mass spectrometer (Q Exactive, Thermo Scientific). Reverse phase separation was performed at a flow rate of 250 nL/min on an EASY-spray Pep-Map RSLC C18 column (75 µM x 500 mm, 2 µm particle size; Thermo Scientific) over a 1 hr gradient of 2% to 35% acetonitrile in 5% DMSO/0.1% Formic Acid. MS1 spectra were acquired with a resolution of 70,000 and an AGC target of 3E6. MSMS spectra were acquired at a resolution of 17,500 for up to 128 ms and an AGC target of 1E5 for up to 15 precursors per duty cycle and a normalised collision energy of 28%. Selected precursors were isolated in the quadrupole with an isolation window of 1.6 m/z and excluded for 7 s for repeated selection. Parallel Reaction Monitoring (PRM) of selected peptides was undertaken on an Orbitrap Fusion Lumos instrument using the same chromatographic parameters as above. iRT peptides (Biognosys) were included as retention time references. MS1 spectra were acquired in the Orbitrap with a resolution of 120,000 and an AGC target of 7E5. PRM scans were acquired in the Orbitrap after quadrupole isolation with 1.2 m/z, HCD fragmentation (25% NCE) and with a resolution of 60,000 in a top 12 duty cycle. AGC target was set to 1E5 and the maximum injection time was 118 ms. PRM scans were scheduled with ±10 min around previously observed retention times.

## Analysis of mass spectrometry data

Processing of DDA data was performed using PEAKS (versions 8.5 and X; Bioinformatics Solutions) with the following parameters applied: 5 ppm mass error tolerance for precursor ion mass and 0.02 Da tolerance for fragment ion masses; protease specificity was semi-specific with up to two missed cleavages; cysteine carbamidomethylation was selected as a fixed modification; variable modifications considered oxidation (C, D, H, F, K, M, N, P, R, W, Y) and di-oxidation (M, W, Y) in addition to deamidation (N, Q) and acetylation (protein N-termini); and a maximum of 3 PTMs per peptide. Additional unspecified PTM searches were considered via Peaks PTM algorithm. Raw data were matched against the canonical human Uniprot reference (retrieved July 9, 2018) supplemented with the modified HIF-1$\alpha$ (M561A/M568A) protein sequence. Peptide false discovery rate was set to 1% using a target/decoy fusion approach. PTM site determination used a probability-based ambiguity score (AScore; $-10$ x log10 P); an AScore value of 20 is equivalent to a p-value of 0.01.

Peptide abundance measurements were obtained by manual peak area integration of extracted ion chromatograms using Qual Browser (Xcalibur; Thermo Scientific); minimal smoothing (Gaussian, three points) was automatically applied to XIC data. Unoxidised non-HIF substrate peptides were all identified by MSMS. For HIF substrate peptides that showed near complete hydroxylation upon enzyme addition, accurate mass and retention time (AMT) signatures were employed, where necessary, to facilitate quantitation of precursor ions in all enzyme-reacted and control samples, including those lacking supporting MSMS. Stringent tolerances were used (5 ppm m/z window and 1 min retention time deviation between related runs) reflecting the reproducible chromatography and high mass accuracy of the nanoLC QExactive platform.

Benchmark deep proteome datasets (PXD002395, PXD003977, PXD004452) were downloaded from the PRIDE repository and batch processed with Peaks 8.5, grouping raw files according to cell type, fragmentation mode and, where appropriate, replicate status. Processing parameters were identical to those used in the in-house DDA data, with the following modifications: (i) precursor mass tolerance set to 10 ppm; (ii) fragment mass tolerance was specific to the MSMS acquisition mode; 0.5 Da for ion trap acquisitions and 0.05 Da for orbitrap; (iii) full tryptic specificity was applied; (iv) data were matched to the unmodified canonical human Uniprot reference (retrieved July 9, 2018); (v) Accurate PTM site determination considered AScore (>20) and minimal relative fragment ion intensity values (major b- or y-fragment ions with >5% relative ion intensity before and after the modified amino acid). Values reported in *Table 3* are indicative of leading peptide scores for each Peaks 8.5 processed dataset, exported as a supporting peptides file (protein-peptides.csv).

Processing of PRM data was performed with Spectrodive 8 (Biognosys) using default analysis settings. Ion chromatograms corresponding to five different fragment ions, which were assigned during spectral library generation (Spectronaut Pulsar X; Biognosys) were extracted with a 10 ppm mass tolerance. Quantitative information (e.g. area counts) and retention time recalibration, facilitated by the use of iRT spike-in peptides in reference (DDA for spectral library generation) and PRM data, was automated in Spectrodive.

## Radioassay for hydroxyproline

PHD substrates were prepared using a HeLa cell lysate-derived IVTT expression system (Thermo Scientific) supplemented with 70 µCi of L-[2,3,4,5-$^3$H]proline (85 mCi/µmol; PerkinElmer Life Sciences) per 25 µl IVTT reaction as described above. The IVTT reaction containing the translated $^3$H-proline labeled substrate was divided into parallel aliquots and subjected to hydroxylation in a reaction volume of 0.5 ml as described above, with each substrate being incubated in parallel with and without the relevant exogenous recombinant PHD. Following the 2 hr hydroxylation reaction at 37°C, the samples were dialysed extensively to remove any remaining free L-[2,3,4,5-$^3$H]-proline, and the 4-[$^3$H]-hydroxyproline formed in the substrate was analyzed by an optimised procedure for the analysis of radiolabelled [$^3$H]-hydroxyproline involving its oxidation to pyrrole (*Juva and Prockop, 1966*). The amount of 4-hydroxy[$^3$H]proline formed during the reaction is given as DPM (disintegrations per minute) 4-hydroxy[$^3$H]proline per $1 \times 10^6$ DPM total [$^3$H]-proline and was calculated by subtracting the DPM values for the reaction carried out without the recombinant PHD. Wild type and P402A/P564G double mutant HIF-1α were included as positive and negative control substrates alongside each hydroxylation assay to ensure efficient hydroxylation of a known substrate and to analyse background DPM generated due to the technical limitations of the assay, respectively.

## Replicates

A summary table detailing the number of independent assays for each target is provided in *Supplementary file 4*.

## Acknowledgements

We thank Eeva Lehtimäki for technical assistance. We thank the Peptide Chemistry and Proteomics Science Technology Platforms of the Francis Crick Institute for provision of reagents and expert advice. Proteomic mass spectrometry analysis was performed at the Discovery Proteomics Facility (headed by Roman Fischer) which is part of the Target Discovery Institute MS Laboratory, University of Oxford (led by Benedikt Kessler). This work was supported by the Francis Crick Institute which receives its core funding from Cancer Research UK (FC001501), the UK Medical Research Council (FC001501), and the Wellcome Trust (FC001501). PJR is also supported by the Ludwig Institute for Cancer Research and the Wellcome Trust (106241/Z/14/Z). CJS, KL, MA, and WDF thank the Wellcome Trust (106244/Z/14/Z), Cancer Research UK, and the British Heart Foundation for funding. KL gratefully acknowledges support via the Newton Abraham D Phil studentship scheme. JM was supported by Academy of Finland project grant 296498, Academy of Finland Center of Excellence 2012–2017 Grant 251314, the S Jusélius Foundation and the Jane and Aatos Erkko Foundation.

## Additional information

### Competing interests

Johanna Myllyharju: equity holder in FibroGen Inc. Christopher J Schofield, Peter J Ratcliffe: scientific co-founder and equity holder in ReOx. The other authors declare that no competing interests exist.

### Funding

| Funder | Grant reference number | Author |
| --- | --- | --- |
| Francis Crick Institute | FC0010501 | Peter J Ratcliffe |

| Ludwig Institute for Cancer Research | | Peter J Ratcliffe |
|---|---|---|
| Wellcome | 106241/Z/14/Z | Peter J Ratcliffe |
| Cancer Research UK | | Christopher J Schofield |
| Wellcome | 106244/Z/14/Z | Christopher J Schofield |
| British Heart Foundation | | Christopher J Schofield |
| Academy of Finland | Project grant 296498 | Johanna Myllyharju |
| Academy of Finland | Center of Excellence 2012-2017 Grant 251314 | Johanna Myllyharju |
| Sigrid Jusélius Foundation | | Johanna Myllyharju |
| Jane and Aatos Erkko Foundation | | Johanna Myllyharju |

The funders had no role in study design, data collection and interpretation, or the decision to submit the work for publication.

## Author contributions

Matthew E Cockman, Conceptualization, Data curation, Supervision, Validation, Visualization, Methodology, Writing—original draft; Kerstin Lippl, Validation, Investigation, Visualization, Writing—review and editing; Ya-Min Tian, Validation, Investigation, Writing—review and editing; Hamish B Pegg, Formal analysis, Investigation, Writing—review and editing; William D Figg Jnr, Formal analysis, Investigation; Martine I Abboud, Resources, Investigation, NMR resources; Raphael Heilig, Resources, Mass spectrometry resources; Roman Fischer, Resources, Methodology, Writing—review and editing, Mass spectrometry resources; Johanna Myllyharju, Christopher J Schofield, Conceptualization, Supervision, Funding acquisition, Writing—review and editing; Peter J Ratcliffe, Conceptualization, Supervision, Funding acquisition, Writing—original draft, Project administration

## Author ORCIDs

Matthew E Cockman (iD) https://orcid.org/0000-0002-3310-4821
William D Figg Jnr (iD) http://orcid.org/0000-0002-5875-2606
Martine I Abboud (iD) http://orcid.org/0000-0003-2141-5988
Roman Fischer (iD) https://orcid.org/0000-0002-9715-5951
Johanna Myllyharju (iD) https://orcid.org/0000-0001-7772-1250
Christopher J Schofield (iD) https://orcid.org/0000-0002-0290-6565
Peter J Ratcliffe (iD) https://orcid.org/0000-0002-2853-806X

## Decision letter and Author response

Decision letter https://doi.org/10.7554/eLife.46490.071
Author response https://doi.org/10.7554/eLife.46490.072

# Additional files

## Supplementary files

- Supplementary file 1. MSMS assignments corresponding to *Figures 2* and *3* and associated supplements.
DOI: https://doi.org/10.7554/eLife.46490.055

- Supplementary file 2. Quantitation of PHD-catalysed prolyl hydroxylation of full-length hypoxia-inducible factor 1α; control hydroxylation assays for HIF-1α corresponding to *Figure 3* and supplements. HIF-1α hydroxylation assays were performed in parallel with alternative substrates to confirm the activity of recombinant PHD enzyme preparations. The relationship between each HIF1A hydroxylation dataset and corresponding alternative substrate assay is indicated above each figure. Full-length HIF-1α protein produced by IVTT was incubated in the absence (Control) or presence of the indicated (PHD) enzyme and digested with trypsin. Endogenous PHD activity in HeLa cell lysate gave rise to a basal (control) level of hydroxylation that markedly increased upon addition of recombinant

PHD. To facilitate quantitation of P564ox we used a variant of HIF1A in which adjacent methionine residues that are prone to de novo oxidation were substituted to alanine (HIF1A*: M561A, M568A). This modified form of HIF1A is more amenable to LC-MS based quantitation (peak intensities are not reduced as a consequence of methionine oxidation) and has no appreciable effect on the rate of PHD-dependent catalysis (*Tian et al., 2011*). Detection of P564 peptide ions was robust across replicate assays and relative quantitation data are presented for all HIF-1α hydroxylation assays. Detection of P402 peptide ions was more variable, requiring an additional (i.e., non-specific) cleavage by trypsin in order to monitor hydroxylation at this site. Relative quantitation of P564ox is presented in panel A, as XIC of m/z 1590.7540 and 1598.7515 corresponding to unoxidised (pink) and P564ox (blue) forms of HIF1A*(548-575). Where present, semi-tryptic peptides encompassing P402 were detected as either HIF1A*(392-403) or HIF1A*(392-411). Panel B shows XIC for precursor ion masses corresponding to unoxidised (pink; (i) m/z 619.8532, (ii) m/z 1005.0619) and P402ox (blue; (i) 627.8506, (ii) 1013.0592) forms of related (i) HIF1A*(392-403) or (ii) HIF1A*(392-411) peptides. Quantitative data for observed species are presented as pie charts with XIC data. Assignment and quantitation data are tabulated below each panel, table headers are as follows: ($-10$lgP) significance score of leading assignment at given (RT) with ambiguity score (AScore) for PTM localisation and reference to primary MSMS data in *Supplementary file 1*. Quantitative data reports ion counts (Area) of observed masses (MH$^{+2}$) integrated over time (RT Win.), expressed as relative abundance (RA).
DOI: https://doi.org/10.7554/eLife.46490.056

• Supplementary file 3. Tabulated inventory of unprocessed MS data (RAW files) associated with IVTT hydroxylation assays presented in *Figure 2* (table i) and *Figure 3* (table ii).
DOI: https://doi.org/10.7554/eLife.46490.057

• Supplementary file 4. Replicate information for peptide hydroxylation assays, IVTT hydroxylation assays and radioassays for hydroxyproline.
DOI: https://doi.org/10.7554/eLife.46490.058

• Transparent reporting form
DOI: https://doi.org/10.7554/eLife.46490.059

## Data availability

All data generated and analysed in this study are included in the manuscript and supporting files. Source data files have been provided for Figures 5 and Table 3. Mass spectrometry raw data files have been uploaded to the ProteomeXchange consortium (LC-MSMS data identifier PXD013112 and http://doi.org/10.6019/PXD013112) and the Dryad repository (LC-MS; http://doi.org/10.5061/dryad.917hb55).

The following datasets were generated:

| Author(s) | Year | Dataset title | Dataset URL | Database and Identifier |
|---|---|---|---|---|
| Cockman ME, Lippl K, Tian Y, Pegg HB, Figg WD, Abboud MI, Heilig R, Fischer R, Myllyharju J, Schofield CJ, Ratcliffe PJ | 2019 | Data from: Lack of activity of recombinant HIF prolyl hydroxylases (PHDs) on reported non-HIF substrates | https://doi.org/10.5061/dryad.917hb55 | Dryad Digital Repository, 10.5061/dryad.917hb55 |
| Cockman ME, Lippl K, Tian Y, Pegg HB, Figg WD, Abboud MI, Heilig R, Fischer R, Myllyharju J, Schofield CJ, Ratcliffe PJ | 2019 | Data from: Lack of activity of recombinant HIF prolyl hydroxylases (PHDs) on reported non-HIF substrates | http://doi.org/10.6019/PXD013112 | ProteomeXchange, 10.6019/PXD013112 |

The following previously published datasets were used:

| Author(s) | Year | Dataset title | Dataset URL | Database and Identifier |
|---|---|---|---|---|
| Geiger T, Wehner A, Schaab C, Cox | 2012 | 11 human cell lines | https://www.ebi.ac.uk/pride/archive/projects/ | PRIDE, PXD002395 |

| | | | | |
|---|---|---|---|---|
| J, Mann M | | | PXD002395 | |
| Davis S, Charles PD, He L, Mowlds P, Kessler BM, Fischer R | 2017 | Expanding proteome coverage with CHarge Ordered Parallel Ion aNalysis (CHOPIN) combined with broad specificity proteolysis | https://www.ebi.ac.uk/pride/archive/projects/PXD003977 | PRIDE, PXD003977 |
| Bekker-Jensen DB, Kelstrup CD, Batth TS, Larsen SC, Haldrup C, Bramsen JB, Sørensen KD, Høyer S, Ørntoft TF, Andersen CL, Nielsen ML, Olsen JV | 2017 | HeLa proteome of 12,250 protein-coding genes | https://www.ebi.ac.uk/pride/archive/projects/PXD004452 | PRIDE, PXD004452 |

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
