## [Decision Letter]

Thank you for submitting your article "Lack of activity of recombinant HIF prolyl hydroxylases (PHDs) on reported non-HIF substrates" for consideration by *eLife*. Your article has been reviewed by three peer reviewers, one of whom is a member of our Board of Reviewing Editors, and the evaluation has been overseen by Utpal Banerjee as the Senior Editor. The reviewers have opted to remain anonymous.

The reviewers have discussed the reviews with one another and the Reviewing Editor has drafted this decision to help you prepare a revised submission. The reviewers in their deliberations are quite clear that they would like to see this work published in *eLife*. The only substantial request is for you to speculate/comment on what might be expected for FIH (see Essential Revisions). The full reviews are included in case you would like to address them, but doing so is entirely discretionary.

Summary:

The discovery of HIF prolyl hydroxylase enzymes raises the important question as to the existence of additional non-HIF substrates, where the HIF-PHDs might govern other biological responses to hypoxia. More than 20 putative substrates have been published previously; however, the authors failed to detect enzymatic activity on any of these peptides or proteins under conditions readily detecting HIFα hydroxylation. While the authors do not exclude prolyl hydroxylation under other conditions not employed in their study, they find no evidence for any of the reported non-HIF substrates in the literature at this time.

Essential revisions:

During the review process, online discussions by the three reviewers have led to a recommendation that the paper include some discussion (in that section of the paper) on the likelihood that the other principle HIFα subunit dioxygenase hydroxylating a C-terminal asparagine residue (FIH) would also have non-HIF substrates, based on your own analyses and assessment of the literature at this time. Moreover, the Discussion section would benefit from added text describing your hypotheses as to why the PHDs are so selective for HIFα subunits, given the noted flexibility of amino acids around their highly conserved prolyl hydroxylation sites. Finally, at least two reviewers recommend several additional experiments that would significantly benefit the paper. In our view these should be able to be completed in approximately two months or less.

Overall, all three reviewers felt a suitably revised manuscript would add significant new information that would benefit the HIF, O_2_-dependent dioxygenase, and oxygen sensing fields.

As such, a revised paper represents a good candidate for the journal. We hope you find the three reviewers' comments provided below helpful and look forward to hearing from you soon.

Title: Suggest "HIF prolyl hydroxylases (PHDs) lack activity toward reported non-HIF substrates".

*Reviewer #1:*

The paper by Cockman et al., entitled "Lack of activity of HIF prolyl hydroxylases (PHDs) on reported non-HIF substrates" represents a technical 'tour de force' that seeks to clarify a large body of literature spanning more than 10 years of work by numerous labs. The background for this important study is that the discovery of the HIF-PHDs as 'oxygen sensors' raised the critical question as to whether there are other prolyl hydroxylation substrates for the three HIF-PHDs (PHD1, PHD2, and PHD3) beyond HIF-α subunits (i.e. α1, α2, and α3). This was an appealing hypothesis, suggesting that non-HIF dependent pathways would similarly be regulated by oxygen availability based on the PHDs as central O_2_ sensors. There are also important clinical implications at play here, in that highly specific PHD enzymatic activity inhibitors are making their way through multiple clinical trials: on target specificity of such drugs for non-HIF substrates could result in significant dose limiting toxicities for patient care. To date, >20 non-HIF PHD substrates have been published. In order to more carefully examine each of these proteins as bone fide enzymatic targets of PHD1-3, the authors carried out exhaustive efforts to measure the prolyl hydroxylation of synthetic peptides (and even full-length purified proteins) representing the reported sites of hydroxylation.

Table 1 summarizes (a) 24 putative non-HIF targets, (b) proposed proline residues in these targets reported to be hydroxylated, (c) which PHD isoform is responsible, and (d) primary references in the literature. To their enormous credit, the authors considered every conceivable technical explanation for why they failed to detect target hydroxylation in their assays (while HIF-1α always served as an internal positive control): (1) length of peptide substrate being inadequate, (2) altered ionization/detection efficiencies in their experiments, (3) co-elution of prolyl hydroxylated species with non-enzymatically oxidized ions, and (4) prolyl hydroxylation that occurs below the limits of detection by their MS approaches (these were explored by independent radiochemical procedures), to name but a few. They also evaluated previously published MS based cellular proteomes for evidence of prolyl hydroxylation of the proposed non-HIF substrates, as an independent unbiased approach. Collectively, they observed no experimental support for the three PHDs having multiple non-HIF substrates, or at least those reported cannot be unequivocally shown to be bona fide PHD substrates.

The authors took great pains to report their findings objectively and comprehensively. As such, this study is a 'landmark' in the O_2_ sensing field and highly worthy of publication expediently as possible, I have no real substantive revisions to suggest, other than expanding the legend for Table 2 to make it easier for the reader to digest without more extensive evaluation of the detailed methods provided in the paper.

*Reviewer #2:*

The discovery of PHD catalyzed hydroxylation of HIF as the key event in the transcriptional response to hypoxia raises the question of whether there are non-HIF substrates that could conceivably enlarge the scope of PHD-regulated hypoxic responses. Over that past decade, over 20 such substrates have been reported. These include protein kinases, transcription factors, metabolic enzymes, and cytoskeletal proteins, among others. The means by which prolyl hydroxylation has been identified in these publications has varied and has included mass spectrometry and anti-hydroxyproline antibodies. In order to independently assess hydroxylation of these reported substrates, the authors have performed an extensive analysis using recombinant PHD1-3, HIF-1α or HIF-2α as a positive control, and three in vitro assays: (1) Incubation of the PHDs with full length substrates from Hela cell extract using vitro transcription/translation (IVTT) reactions followed by immunopurification and mass spec; (2) incubation of the PHDs with substrate derived peptides followed by mass spec; (3) incubation of the PHDs with IVTT proteins in the presence of 3H-proline, followed by assessment of hydroxylation of 3H-proline. The surprising result is that under conditions in which HIF hydroxylation is readily detected, they did not find evidence of hydroxylation of any of the non-HIF substrates reported in the literature.

The experiments are well performed, systematic, and meticulously reported. The data are convincing. The use of independent assays and the inclusion of HIF positive controls in these assays lends substantial weight to the results. The author appropriately discuss limitations of their study, which include the possibility that hydroxylation might occur in a cellular environment but not in vitro. Their findings are at odds with a large number of publications that report non-HIF substrates, and as such, would represent a very important contribution to the literature. Undoubtedly, this will provoke further discussion and investigation.

*Reviewer #3:*

This article describes a very thorough investigation of the hydroxylation of essentially all of the reported non-HIF substrates of the PHD enzymes. The use of both peptides and full length proteins, exhaustive analysis by mass spectrometry, including determining of differences in ionisation efficiency between hydroxylated and non-hydroxylated peptides, and the complementary methodology of radiolabeled enzyme assays is a major strength of this study. The results are generally very well presented and described, although complex and not necessarily appreciated by a more general audience outside this specific field. The major conclusions drawn from the in vitro data are very well supported.

The results are unexpected, given the numerous publications presenting both cell-based and in vitro evidence by mass spectrometry for hydroxylation of these substrates. These findings make a very important contribution to this field, and raise doubt as to whether any of these are actually substrates in vivo, and if they are why they do not appear to be substrates in vitro. This, of course, has important implications for the therapeutic use of PHD-specific inhibitors.

However, with only the in vitro experiments included in this study, the key question of whether any of these are substrates in vivo is not addressed, at least experimentally. The authors do propose a number of potential reasons for this lack of hydroxylation in vitro compared to reports in cells, which are logical, from none of these being substrates of the PHDs (artefacts), through to the lack of necessary cellular components in the in vitro assay, or assay-specific differences.

What is lacking from this study is some attempt to rationalise this key issue, by attempting to replicate some of the published cell-based experiments using the more the intensive interrogation by mass spectrometry used in this study. This analysis could have been performed, for example, by hydroxylating substrates in vitro using whole cell extracts, with PHDs either inhibited, knocked down or knocked out, as used in other studies, to determine whether additional factors are required. Or by extracting and analysing endogenous proteins to determine whether it is more likely an issue with interpretation of the mass spectrometry such as misinterpretation of oxidation. Ultimately, we are left with a set of very provocative results, but uncertain whether this is an artefact of the in vitro system or of real physiological significance.

---

## [Author Response]

Essential revisions:During the review process, online discussions by the three reviewers have led to a recommendation that the paper include some discussion (in that section of the paper) on the likelihood that the other principle HIFα subunit dioxygenase hydroxylating a C-terminal asparagine residue (FIH) would also have non-HIF substrates, based on your own analyses and assessment of the literature at this time. Moreover, the Discussion section would benefit from added text describing your hypotheses as to why the PHDs are so selective for HIFα subunits, given the noted flexibility of amino acids around their highly conserved prolyl hydroxylation sites. Finally, at least two reviewers recommend several additional experiments that would significantly benefit the paper. In our view these should be able to be completed in approximately two months or less.Overall, all three reviewers felt a suitably revised manuscript would add significant new information that would benefit the HIF, O_2_-dependent dioxygenase, and oxygen sensing fields.As such, a revised paper represents a good candidate for the journal. We hope you find the three reviewers' comments provided below helpful and look forward to hearing from you soon.Title: Suggest "HIF prolyl hydroxylases (PHDs) lack activity toward reported non-HIF substrates".

The reviewing editor has asked that we add text to the discussion concerning why the PHDs are apparently so selective for their HIF-α substrates. This question is indeed of interest, because, as the editor points out, many amino acid substitutions can be tolerated in HIF-α peptides without the total loss of activity that was observed with the non-HIF substrates tested in these experiments.

We are not yet sure of the precise reasons, which are the subject of on-going work. However, X-ray and NMR structural analyses of HIF-α peptides in complex with both human PHD2 protein and that of the simplest animal *Trichoplax adhaerens* reveal that the bound peptide makes multiple contacts with PHD2. There are also substantial conformational changes in the PHD structures on binding HIF-α substrates, including those involving at least one loop, which on substrate binding moves to enclose the catalytic site and the C-terminal region of the enzyme(1-3). Thus (so far), the complex nature of the protein-protein interactions between the PHDs and their HIF substrates means it is difficult to predict PHD substrates from primary sequences (as is the case for other 2-oxoglutarate dependent oxygenases (4).

In revision, we have added a brief discussion of this. We have also (as suggested by reviewer 2) added a supplementary table comparing both predicted and defined secondary structure in HIF-α peptides and the reported non-HIF substrates. This reveals that, despite substantial primary sequence variation, many (possibly all) HIF-α peptides are predicted to form a helix immediately adjacent to the target prolyl residue. This prediction is supported by crystal structures of PHD2 in complex with CODD and NODD. However, a *caveat* is that binding to the enzyme may impose other structural changes and constraints, which cannot be predicted from this type of bioinformatic analysis.

The reviewing editor also asks that we discuss the likelihood that the HIF asparaginyl hydroxylase, FIH has other substrates. We (and others) have reported that FIH hydroxylates asparaginyl (and other) residues in a wide range of ankyrin-repeat domain (ARD)-containing proteins, in addition to HIF-α. As an additional control for this study we tested these substrates in the IVTT assays with FIH and confirmed that both ARD-containing proteins and HIF-α are efficiently hydroxylated by FIH. Multiple X-ray structures of peptide substrates complexed to FIH imply a much more open binding site than PHD2/HIF-α, with less induced fit (at least from the enzyme perspective), with FIH substrate peptides binding in a largely extended configuration(5, 6). These results, together with the observation that suitably sited residues in the ankyrin fold other than asparagine may be hydroxylated(7), suggests that the substrate specificity of FIH may be much broader than the PHDs. However, FIH must unfold its ankyrin repeat substrates from their canonical structure and the extent to which this process constrains the FIH substrate repertoire is not understood. Nevertheless, we agree the subject is of interest and have added text covering this to the Discussion section of the revised manuscript.

Reviewer #1:[…]The authors took great pains to report their findings objectively and comprehensively. As such, this study is a 'landmark' in the O_2_ sensing field and highly worthy of publication expediently as possible, I have no real substantive revisions to suggest, other than expanding the legend for Table 2 to make it easier for the reader to digest without more extensive evaluation of the detailed methods provided in the paper.

We have expanded the legend to Table 2 as suggested.

Reviewer #2:The discovery of PHD catalyzed hydroxylation of HIF as the key event in the transcriptional response to hypoxia raises the question of whether there are non-HIF substrates that could conceivably enlarge the scope of PHD-regulated hypoxic responses. Over that past decade, over 20 such substrates have been reported. These include protein kinases, transcription factors, metabolic enzymes, and cytoskeletal proteins, among others. The means by which prolyl hydroxylation has been identified in these publications has varied and has included mass spectrometry and anti-hydroxyproline antibodies. In order to independently assess hydroxylation of these reported substrates, the authors have performed an extensive analysis using recombinant PHD1-3, HIF-1α or HIF-2α as a positive control, and three in vitro assays: (1) Incubation of the PHDs with full length substrates from Hela cell extract using vitro transcription/translation (IVTT) reactions followed by immunopurification and mass spec; (2) incubation of the PHDs with substrate derived peptides followed by mass spec; (3) incubation of the PHDs with IVTT proteins in the presence of 3H-proline, followed by assessment of hydroxylation of 3H-proline. The surprising result is that under conditions in which HIF hydroxylation is readily detected, they did not find evidence of hydroxylation of any of the non-HIF substrates reported in the literature.

Point 1: Like the reviewing editor, this reviewer also raises the question of substrates specificity determinants for the PHDs. We agree this is an interesting point; our response is as set out above.

Point 2: We were uncertain what was intended in this comment, as the effect of prolyl hydroxylation on the chromatographic behaviour of the peptide standards is summarised at the point of reference to the tabulated data (Table 2). This can be found towards the end of the Results section. It could that the reviewer was suggesting that we give this data earlier but overall we think that the flow of the manuscript is best with the text as written, which is in line with the order in which the work was performed.

Point 3: We have added further detail on the radiochemical assay. We have also revised Figure 5 in a way which we think more accurately portrays the data, in particular we have extended the scale below zero to indicate that values for the non-HIF substrates include negative values. These values cluster around zero, rather than the small positive value that might have been suggested by the way the figure was originally drawn. The IVTT reaction containing the translated [^3^H]-proline labeled substrate was divided into two aliquots and subjected to hydroxylation with and without the recombinant PHD. The data is now given as DPM [^3^H]-hydroxyproline formed during the reaction per 1 x 10^6^ DPM total [^3^H]-proline. This is calculated by subtracting such DPM values for the control reaction carried out without the recombinant PHD from the test reaction. Wild-type and the P402A/P564G double mutant HIF-1α are included as positive and negative control substrates in each hydroxylation assay, to verify efficient hydroxylation of a known substrate and to display the background DPM range due to technical limitations of the assay, respectively. As requested, details are provided in the revised Materials and methods section.

The experiments are well performed, systematic, and meticulously reported. The data are convincing. The use of independent assays and the inclusion of HIF positive controls in these assays lends substantial weight to the results. The author appropriately discuss limitations of their study, which include the possibility that hydroxylation might occur in a cellular environment but not in vitro. Their findings are at odds with a large number of publications that report non-HIF substrates, and as such, would represent a very important contribution to the literature. Undoubtedly, this will provoke further discussion and investigation.

The reviewer raises two questions. First, based on existing evidence, are the proposed non-HIF substrates unstructured in the regions containing the proposed site of hydroxylation? Second, what is the extent of conservation across metazoans in the regions of HIF-α polypeptides that are targeted for prolyl hydroxylation.

These questions impinge on the editor’s suggestion that we discuss hypotheses as to the determinants of PHD selectivity towards HIF substrates. As outlined above our response to the editor, we have added a supplementary table comparing both predicted and defined secondary structure in HIF-α regions that are targeted for prolyl hydroxylation in different metazoan species, together with similar analyses on reported non-HIF substrates. This reveals that, despite substantial primary sequence variation, HIF-α peptides are predicted to form an α-helix close to the target prolyl residue. In contrast, we observed no common secondary structure (either observed or predicted) in the region of the proposed sites of prolyl hydroxylation in reported non-HIF substrates.

Reviewer #3:This article describes a very thorough investigation of the hydroxylation of essentially all of the reported non-HIF substrates of the PHD enzymes. The use of both peptides and full length proteins, exhaustive analysis by mass spectrometry, including determining of differences in ionisation efficiency between hydroxylated and non-hydroxylated peptides, and the complementary methodology of radiolabeled enzyme assays is a major strength of this study. The results are generally very well presented and described, although complex and not necessarily appreciated by a more general audience outside this specific field. The major conclusions drawn from the in vitro data are very well supported.The results are unexpected, given the numerous publications presenting both cell-based and in vitro evidence by mass spectrometry for hydroxylation of these substrates. These findings make a very important contribution to this field, and raise doubt as to whether any of these are actually substrates in vivo, and if they are why they do not appear to be substrates in vitro. This, of course, has important implications for the therapeutic use of PHD-specific inhibitors.However, with only the in vitro experiments included in this study, the key question of whether any of these are substrates in vivo is not addressed, at least experimentally. The authors do propose a number of potential reasons for this lack of hydroxylation in vitro compared to reports in cells, which are logical, from none of these being substrates of the PHDs (artefacts), through to the lack of necessary cellular components in the in vitro assay, or assay-specific differences.What is lacking from this study is some attempt to rationalise this key issue, by attempting to replicate some of the published cell-based experiments using the more the intensive interrogation by mass spectrometry used in this study. This analysis could have been performed, for example, by hydroxylating substrates in vitro using whole cell extracts, with PHDs either inhibited, knocked down or knocked out, as used in other studies, to determine whether additional factors are required. Or by extracting and analysing endogenous proteins to determine whether it is more likely an issue with interpretation of the mass spectrometry such as misinterpretation of oxidation. Ultimately we are left with a set of very provocative results, but uncertain whether this is an artefact of the in vitro system or of real physiological significance.

The reviewer raises an issue as to whether we should attempt further experiments to resolve the differences between the results of the work we report, and those reporting non-HIF substrates. In particular, it is suggested that we might attempt to hydroxylate substrates in vitro using whole cell extracts, or that we extract and analyse endogenous proteins.

First, we should point out that our IVTT experiments used substrates that were produced in Hela cell extracts. In all reactions conducted with HIF-α polypeptides as positive controls, we observed an increase in hydroxylation of the anticipated prolyl residues upon addition of the relevant recombinant PHD. However, we also observed clear evidence of prolyl hydroxylation at the HIF target sites in IVTT samples prior to these reactions. Thus, if endogenous PHD activity in whole cell extracts were sufficient to hydroxylate the reported non-HIF substrates then this should have been evident in the analyses that we report. We cannot exclude that some cell type difference (e.g. in PHD isoform expression) that was not captured in our experiments might nevertheless be capable of supporting hydroxylation, which is why we are careful to point out that we cannot exclude hydroxylation occurring under conditions other than those of our experiments.

The reviewer is correct that in theory it would be possible to extract and analyse endogenous proteins, at least those for which efficient immuno-precipitating antibodies exist. However, together with the necessary analytical controls, and interventions on enzyme activity, this would entail a very extensive further set of technically challenging experiments (especially at natural abundance levels), well beyond work which would be consistent with timely revision. We also think that it is important to publish at this stage in an unprejudiced manner, so that others are aware of the results in planning their research and have the opportunity to contribute further.

References

1) Chowdhury R, McDonough MA, Mecinovic J, Loenarz C, Flashman E, Hewitson KS, et al. Structural basis for binding of hypoxia-inducible factor to the oxygen-sensing prolyl hydroxylases. Structure. 2009;17(7):981-9.

2) Lippl K, Boleininger A, McDonough MA, Abboud MI, Tarhonskaya H, Chowdhury R, et al. Born to sense: biophysical analyses of the oxygen sensing prolyl hydroxylase from the simplest animal Trichoplax adhaerens. Hypoxia. 2018;6:57-71.

3) Chowdhury R, Leung IK, Tian YM, Abboud MI, Ge W, Domene C, et al. Structural basis for oxygen degradation domain selectivity of the HIF prolyl hydroxylases. Nature communications. 2016;7:12673.

4) Markolovic S, Leissing TM, Chowdhury R, Wilkins SE, Lu X, Schofield CJ. Structure-function relationships of human JmjC oxygenases-demethylases versus hydroxylases. Curr Opin Struct Biol. 2016;41:62-72.

5) Elkins JM, Hewitson KS, McNeill LA, Seibel JF, Schlemminger I, Pugh CW, et al. Structure of factor-inhibiting hypoxia-inducible factor (HIF) reveals mechanism of oxidative modification of HIF-1 α. Journal of Biological Chemistry. 2003;278(3):1802-6.

6) Coleman ML, McDonough MA, Hewitson KS, Coles C, Mecinovic J, Edelmann M, et al. Asparaginyl hydroxylation of the Notch ankyrin repeat domain by factor inhibiting hypoxia-inducible factor. Journal of Biological Chemistry. 2007;282(33):24027-38.

7) Yang M CR, Ge W, Hamed R, McDonough M, Claridge T, Kessler B, Cockman M, Ratcliffe PJ and Schofield CJ. Factor-Inhibiting Hypoxia-Inducible Factor (FIH) Catalyses the Post-translational Hydroxylation of Histidinyl Residues within Ankyrin Repeat Domains. Febs Journal. 2011;278(7):1086-97.